# Genome sequencing unveils *bla*KPC-2-harboring plasmids as drivers of enhanced resistance and virulence in nosocomial *Klebsiella pneumoniae*

Xinhong Han,[1,2,3] Junxin Zhou,[1,2,3] Lifei Yu,[4] Lina Shao,[1] Shiqi Cai,[1,2,3] Huangdu Hu,[1,2,3] Qiucheng Shi,[1,2,3] Zhengan Wang,[1,2,3] Xiaoting Hua,[1,2,3] Yan Jiang,[1,2,3] Yunsong Yu[1,2,3]

**ABSTRACT**  The threat posed by *Klebsiella pneumoniae* in healthcare settings has worsened due to the evolutionary advantages conferred by *bla*KPC-2-harboring plasmids (pKPC-2). However, the specific evolutionary pathway of nosocomial *K. pneumoniae* carrying pKPC-2 and its transmission between patients and healthcare environments are not yet well understood. Between 1 August and 31 December 2019, 237 ST11 KPC-2-producing-carbapenem-resistant *K. pneumoniae* (CRKP) (KPC-2-CRKP) were collected from patient or ward environments in an intensive care unit and subjected to Illumina sequencing, of which 32 strains were additionally selected for Nanopore sequencing to obtain complete plasmid sequences. Bioinformatics analysis, conjugation experiments, antimicrobial susceptibility tests, and virulence assays were performed to identify the evolutionary characteristics of pKPC-2. The pKPC-2 plasmids were divided into three subgroups with distinct evolutionary events, including Tn*3*-mediated plasmid homologous recombination, IS*26*-mediated horizontal gene transfer, and dynamic duplications of antibiotic resistance genes (ARGs). Surprisingly, the incidence rates of multicopy *bla*KPC-2, *bla*SHV-12, and *bla*CTX-M-65 were quite high (ranging from 27.43% to 67.01%), and strains negative for extended-spectrum β-lactamase tended to develop multicopy *bla*KPC-2. Notably, the presence of multicopy *bla*SHV-12 reduced sensitivity to ceftazidime/avibactam (CZA), and the relative expression level of *bla*SHV-12 in the CZA-resistant group was 6.12 times higher than that in the sensitive group. Furthermore, a novel hybrid pKPC-2 was identified, presenting enhanced virulence levels and decreased susceptibility to CZA. This study emphasizes the notable prevalence of multicopy ARGs and provides a comprehensive insight into the intricate and diverse evolutionary pathways of resistant plasmids that disseminate among patients and healthcare environments.

**IMPORTANCE**  This study is based on a CRKP screening program between patients and ward environments in an intensive care unit, describing the pKPC-2 (*bla*KPC-2-harboring plasmids) population structure and evolutionary characteristics in clinical settings. Long-read sequencing was performed in genetically closely related strains, enabling the high-resolution analysis of evolution pathway between or within pKPC-2 subgroups. We revealed the extremely high rates of multicopy antibiotic resistance genes (ARGs) in clinical settings and its effect on resistance profile toward novel β-lactam/β-lactamase inhibitor combinations, which belongs to the last line treatment choices toward CRKP infection. A novel hybrid pKPC-2 carrying CRKP with enhanced resistance and virulence level was captured during its clonal spread between patients and ward environment. These evidences highlight the threat of pKPC-2 to CRKP treatment and control. Thus, surveillance and timely disinfection in clinical settings should be practiced to prevent transmission of CRKP carrying threatful pKPC-2. And rational use of antibiotics should be called for to prevent inducing of pKPC-2 evolution, especially the multicopy ARGs.

Address correspondence to Yan Jiang, jiangy@zju.edu.cn, or Yunsong Yu, yvys119@zju.edu.cn.

Xinhong Han and Junxin Zhou contributed equally to this article. Author order was determined by type of contribution.

The authors declare no conflict of interest.

See the funding table on p. 14.

**KEYWORDS** carbapenem-resistant *K. pneumoniae*, KPC-2 carbapenemase, long-read sequencing, plasmid evolution, novel β-lactam/β-lactamase inhibitor combinations

The threat of *Klebsiella pneumoniae*, which has become one of the most problematic pathogens in hospitals, has accelerated due to horizontal gene transfer (HGT), especially of carbapenem resistance genes via mobilizable plasmids (1). The transfer of carbapenem resistance genes, such as $bla_{KPC-2}$, $bla_{NDM-1}$, and $bla_{OXA-48}$, contributes to the rise in β-lactam resistance, which further results in there being limited treatment options for clinical *K. pneumoniae* infections; thus, increasing the likelihood of poor clinical outcomes (2).

*K. pneumoniae* carbapenemase (KPC), a class A serine β-lactamase, is prevalent worldwide, and KPC-2 is the predominant carbapenemase in China (3). The international pandemic of KPC-2 is primarily associated with a successful clonal group 258 (CG258), which includes ST258 and ST11 (4). $bla_{KPC-2}$ is located on plasmids and transmits not only vertically within the clonal lineage, but also horizontally between strains, resulting in diversity within $bla_{KPC-2}$-harboring plasmids (pKPC-2) (5). Distinct pKPC-2 replicons have been reported, such as IncFII, IncFIB, IncR, IncI2, IncX, and ColE1 (6, 7). Meanwhile, mobile genetic elements (MGEs), including insertion sequences, transposons, and gene integrons, often have multiple copies at different locations on plasmids (8). With the development of short-read whole-genome sequencing (WGS), it was identified that the genetic environment around the $bla_{KPC-2}$ gene was a Tn*3* family transposon, including Tn*4401*, Tn*1721*, and their variants, which allowed the $bla_{KPC-2}$ gene to mobilize among multiple plasmids (7, 9). Although pKPC-2 plasmids are complex and threatening, studies limit the focus on the population structure or the evolution of pKPC-2s when analyzing the clonally evolving KPC-2-CRKP lineage, likely because of the limitations of short-read WGS, which interrupts the complete and accurate structure formation of plasmids (5).

Using long-read sequencing, plasmid recombination and rearrangement were sporadically observed, and it was determined that plasmid-mediated evolution could accelerate bacterial adaptation to the environment (8, 10). For instance, Jin et al. reported a hybrid pKPC-2 plasmid simultaneously conferring carbapenem resistance and enhanced virulence in a clinical CRKP strain with no significant fitness cost (11). In one experimental study, mutations within $bla_{KPC-2}$ were reported when the bacteria were under the selective pressure of CAZ/AVI, which resulted in a high level of CAZ/AVI resistance (12). Due to the critical condition of patients, frequent invasive operations, and increased use of antibacterial drugs, the intensive care unit (ICU) is considered an important source of CRKP infection, and its isolates tend to evolve faster when under heavy antibiotic selective pressure. However, with the widely distributed and rapid changes of MGEs, it is difficult to distinguish the specific evolutionary process related to pKPC-2 from its chaotic genetic structure. As a result, few studies have systematically analyzed the genetic diversity of the pKPC-2 population from clinical endemic CRKP clones (13).

From 1 August to 31 December 2019, we collected 237 ST11 KPC-2-CRKP isolates from a 22-week ICU CRKP screening program and aimed to track the high-resolution map of evolution for pKPC-2 plasmids. To analyze the detailed evolutionary process, 32 homologous CRKP isolates in which the pKPC-2 plasmids changed at a distinguishable pace were selected for both long- and short-read sequencing to resolve the complete and accurate plasmid structure. Subsequent phylogenetic analysis and BLASTN were performed to investigate the evolutionary events that altered pKPC-2. Then, a collection of short-read sequence data of the 237 ST11 KPC-2-CRKP isolates was used to estimate the frequency of pKPC-2 plasmid evolution events. Conjugation, antimicrobial susceptibility testing, and virulence assays were performed to investigate the effect of pKPC-2 evolution on its host bacteria.

## RESULTS

### Isolation and genomic characteristics of the KPC-2-CRKP isolates

During the 22-week CRKP screening program, a total of 237 ST11 KPC-2-CRKP were detected from the 8,668 samples collected in the 28-bed ICU. According to cgMLST analysis, the strains were divided into three clusters, among which cluster 1 contained 225 serotype K-locus (KL) 64 CRKP and 1 KL64-like CRKP, cluster 2 contained 8 KL105 CRKP, and cluster 3 contained 3 KL19 CRKP (Fig. S1).

The eighth week had the highest ST11 KPC-2-CRKP isolation number, and the 32 CRKP isolates were widely disseminated between patients and their respective environments in the ICU (Fig. 1). WGS analysis suggested that the 32 CRKP isolates all belonged to KL64 and were extremely close, with 0 to 40 Single-Nucleotide Polymorphism (SNP) differences, which were then selected for long-read sequencing and tracking of the high-resolution $bla_{KPC-2}$-harboring plasmid (pKPC-2 plasmid) evolutionary pathway during the CRKP clonal spread process (Fig. 1).

### Population structure and phylogenetic analysis of $bla_{KPC-2}$-harboring plasmids

To investigate the evolutionary characteristics of pKPC-2 plasmids, phylogenetic analysis of all 32 pKPC-2 plasmids was performed. Here, all replicons of pKPC-2 belonged to the IncF type, and these were further divided into three subgroups: small-size (16/32, 70 kb to 92 kb) IncFII(pHN7A8)/IncR-type plasmids, large-size (13/32, 133 kb to 141 kb) IncFII(pHN7A8)/IncR-type plasmids, and IncHI1B/IncFIB(K)-type plasmids (3/32, 144 kb to 254 kb) (Fig. 2). For IncFII/IncR-type pKPC-2 plasmids, they had identical structures in each subgroup, except for the difference in antibiotic resistance gene (ARG) copy number, which included $bla_{KPC-2}$ in subgroup 1 and $bla_{CTX-M-65}$ in subgroup 2 (Fig. S2a/b). The $bla_{KPC-2}$ gene was the only ARG harbored by pKPC-2 plasmids in subgroup 1. In contrast, there were two additional regions on pKPC-2 plasmids in subgroup 2, including a multidrug-resistance region ($bla_{SHV-12}$, $rmtB$, $bla_{TEM-1}$, and $bla_{CTX-M-65}$) and a metal resistance region (Fig. S2c). The three IncHI1B/IncFIB(K)-type pKPC-2 plasmids were all isolated from Bed unit 16, among which virulence gene clusters ($rmpA2$ and $iutAiu-cABCD$) and multicopy $bla_{SHV-12}$ were observed (Fig. 3a).

### Plasmid recombination and HGT events in the evolution process of pKPC-2 in the nosocomial environment

#### Recombination traces between two IncFII/IncR subgroup pKPC-2 plasmids

The two subgroups of IncFII/IncR-type plasmids exhibited similar skeletons with evidence of recombination events. Compared to pKPC-2 in subgroup 1, plasmids in subgroup 2 carried two additional regions. The metal resistance region was flanked by IS5075 and a 28-bp specific recombination site, which was reported to exist in 31.8% of the conjugative plasmids and 98.8% of the virulence plasmids; furthermore, it is a marker of site-specific recombination (14). The antibiotic resistance region (encoding $bla_{SHV-12}$, $bla_{TEM-1}$, $bla_{CTX-M-65}$, and $rmtB$) was flanked by the 8-bp target site (GTGCAGCT) of IS26 (Fig. S2c).

#### Formation of the novel hybrid IncHI1B/IncFIB(K)-type pKPC-2 via virulence and carbapenem resistance plasmids during clonal spread of CRKP

The genetic structure of pKB16_E2_KPC in KB16_E2 was almost identical to that of the IncHI1B/IncFIB(K)-type virulence-encoding plasmid in KB16_E1 (pKB16_E1_viru), except for the two additional resistance-encoding segments ($bla_{KPC-2}$ and multicopy $bla_{SHV-12}$-encoding segment) which were identical to the regions in the pKPC-2 plasmid in KB16_E1. This presented the possibility that the hybrid pKPC-2 was formed through fusion of the virulence plasmid and the pKPC-2 plasmid in CRKP isolates from the same bed unit.

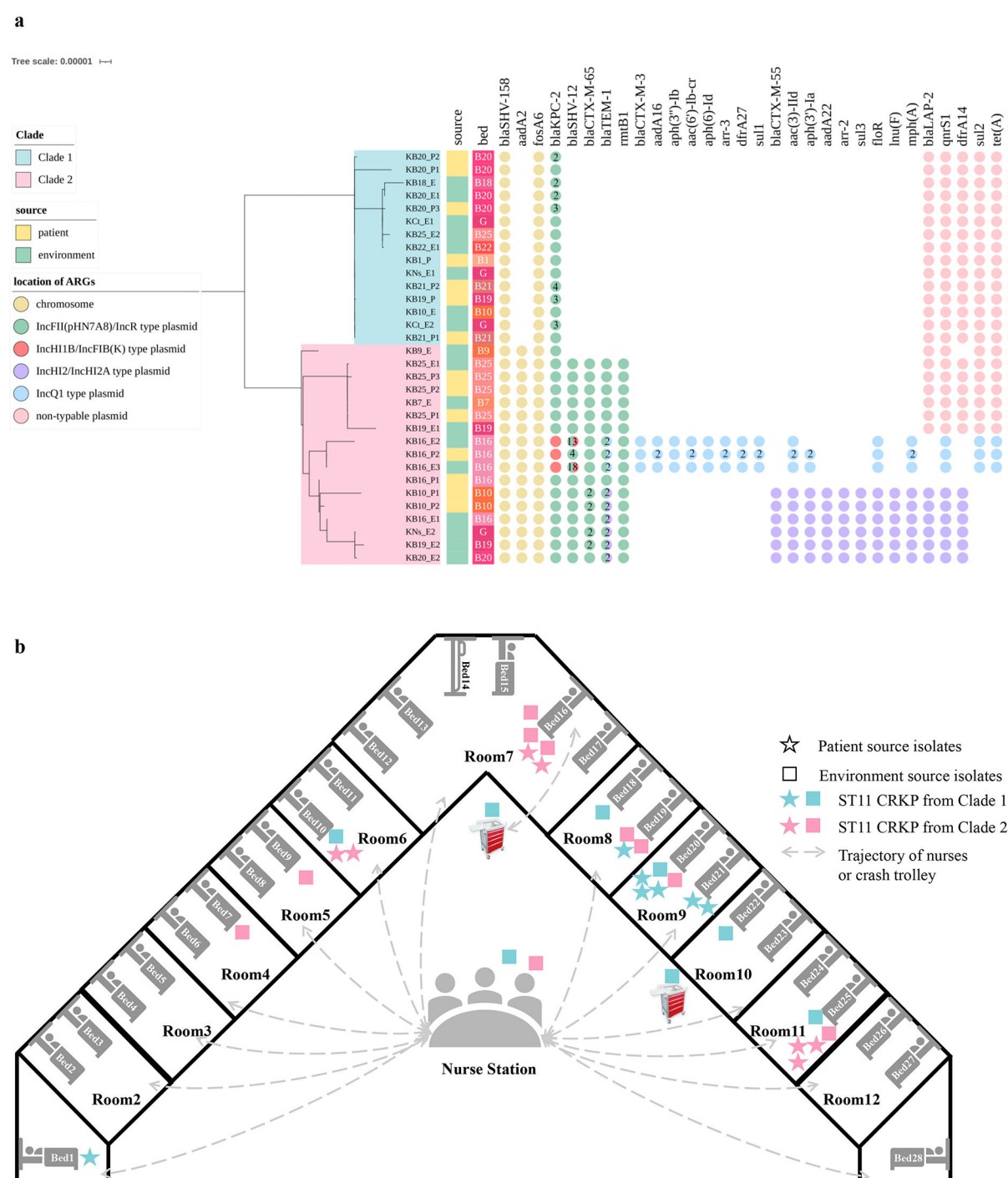

**FIG 1** Phylogenetic analysis and distribution of the 32 ST11 CRKP isolates. (a) Core-genome phylogenetic tree of ST11 CRKP isolates generated by Roary 3.13.0 and IQ-TREE 2.0.3, including information about isolate source, bed unit, and antibiotic resistance genes (ARGs) with location. The numbers on the circles represent the copy number of ARGs. (b) The distribution of CRKP isolates in the ICU. The stars and squares represent CRKP isolated from patients and environmental sites, respectively. The CRKP isolates were divided into different color groups according to the phylogenetic tree. The trajectory of nurses or crash trolley is indicated by a dashed line with an arrow. According to medical records, the crash trolley was only used once in Bed 16 in the past 2 weeks.

Further sequencing analysis confirmed that the hybrid pKB16_E2_KPC was generated via a combination of pKB16_E1_KPC and pKB16_E1_viru through two steps, including HGT mediated by IS*26* insertion and homologous recombination associated with the Tn*3* family transposons (Fig. 3b). Specifically, for the *bla*$_{SHV-12}$-encoding region,

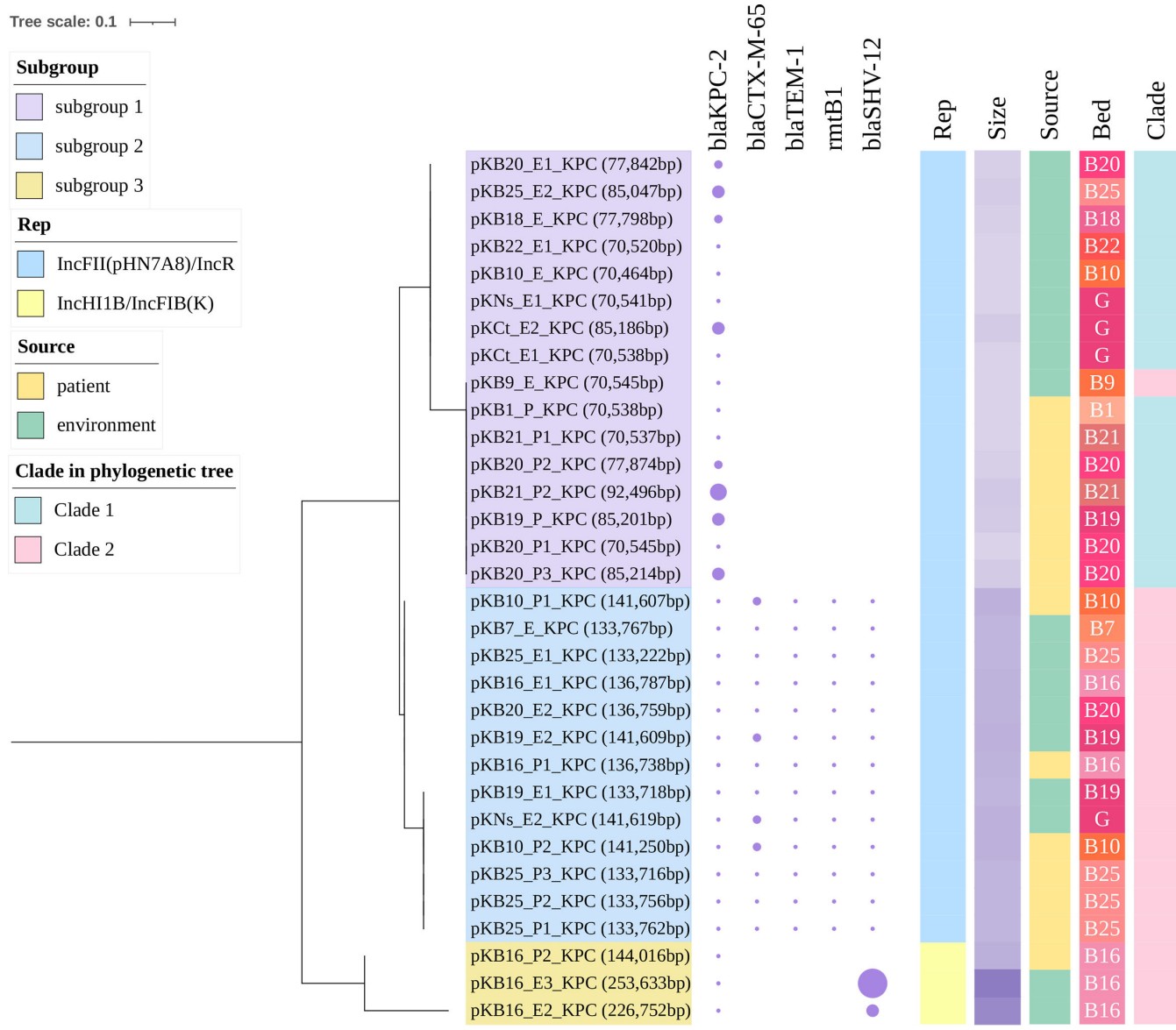

**FIG 2** Subtyping of the 32 pKPC-2 plasmids in the study. The phylogenetic tree was generated by MAFFT 7.310 and IQ-TREE 2.0.3, including information on pKPC-2 plasmid-borne ARGs, plasmid replicons, plasmid size, and plasmid host CRKP isolates. The size of the circle represents the relative copy number of ARGs.

pKB16_E1_viru contained the same 8-bp IS*26*-targeted sites as pKB16_E1_KPC at both 39,455 bp and 39,462 bp (CTTTATCG). The translocatable *bla*$_{SHV-12}$-carrying IS*26* unit was excised from pKB16_E1_KPC and then inserted into the target site on pKB16_E1_viru with an inversion, replacing the segment between the target sites on pKB16_E1_viru. For the *bla*$_{KPC-2}$-encoding region, homologous recombination occurred between the *bla*$_{KPC-2}$-encoding segment (49,030 bp to 92,056 bp) on pKB16_E1_KPC and the 169,568 bp to 180,342 bp segment on pKB16_E1_viru. In the left recombination fragment, Tn*As1* and Tn*As2*, both of which belong to the Tn*3* family transposons, were present on the two plasmids, and the 100 bp homology segment was 100% coverage and 84% identity. In the right recombination fragment, the homologous region, containing the 14-bp identical sequence in both pKB16_E1_viru and pKB16_E1_KPC, extended to 252 bp with 83.14% nucleotide identity.

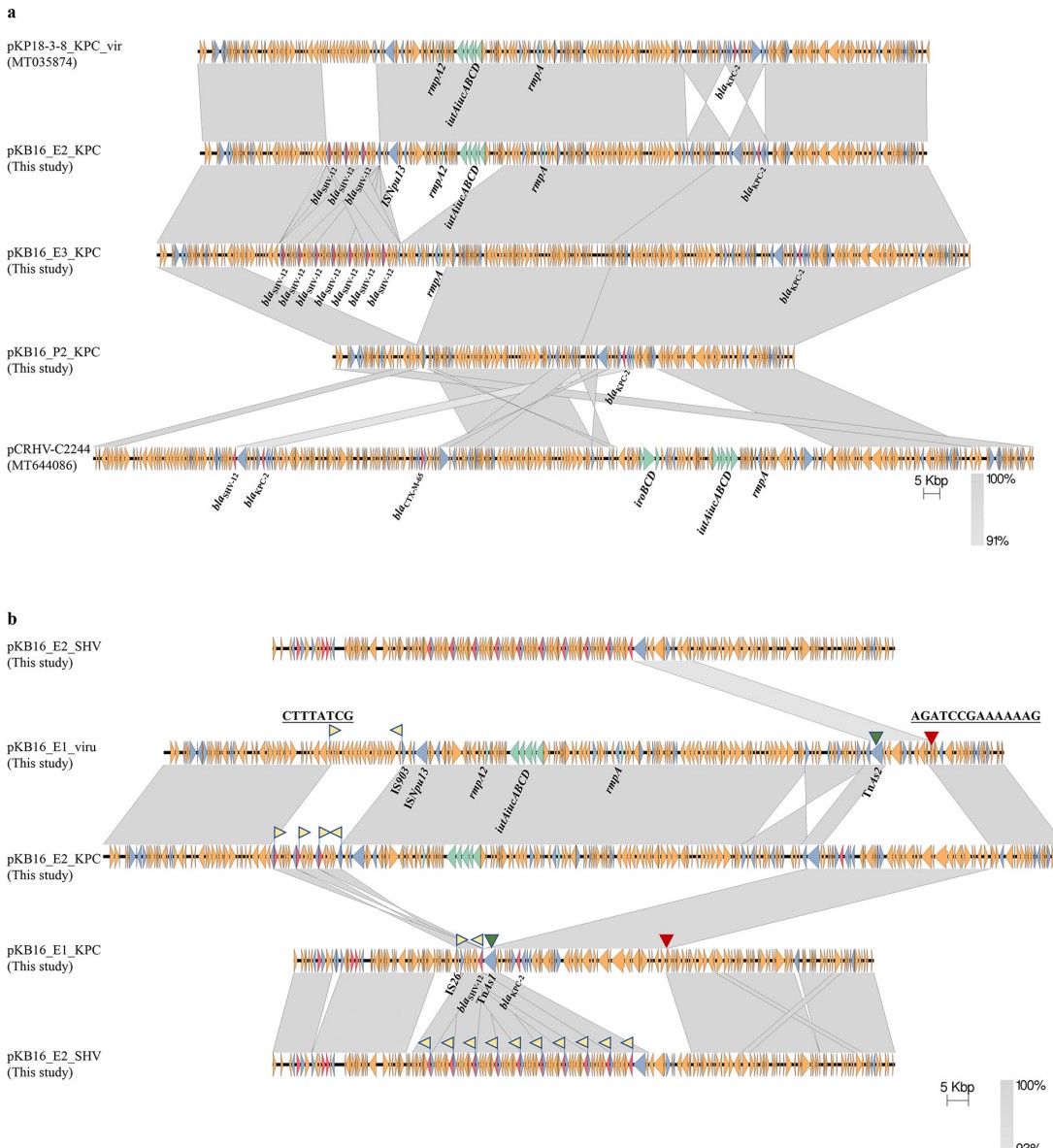

FIG 3 Structure and formation of the hybrid pKB16_E2_KPC plasmid. (a) Linear comparisons of the IncHI1B/IncFIB(K)-type pKPC-2 plasmids with other plasmids that have homologous structures. Three IncHI1B/IncFIB(K)-type pKPC-2 plasmids were aligned with each other and were compared with two virulence-resistance-harboring plasmids. pKB16_E2_KPC shared the most similar genetic backbone with pCRHV-C2244 (GenBank accession no. MT644086) and pKP18-3-8_KPC_vir (GenBank accession no. MT035874) (with 95% and 94% coverage, and 100% and 99.93% identity, respectively). (b) The hybrid plasmid pKB16_E2_KPC was fused through recombination of pKB16_E1_viru and pKB16_E1_KPC via IS26-mediated HGT and Tn3 family-mediated homologous recombination. The yellow flag represents the 8-bp target site of IS26, the green triangle represents the left homologous segment, and the red triangle represents the right homologous segment.

## Amplification of ARGs in the evolution of pKPC-2 in the nosocomial environment

### Amplification of ARGs occurred quite frequently

A total of 237 ST11 KPC-2-CRKP isolates were tested to understand the frequency of the amplification phenomenon observed above (Table S1). The estimated ARG copy number suggested that amplification of ARGs was common in ST11 KPC-2-CRKP isolates from the ICU. Specifically, the estimated $bla_{KPC-2}$ copy number in 27.43% of the KPC-2-CRKP isolates was more than two. Among the 97 $bla_{SHV-12}$-positive KPC-2-CRKP isolates,

67.01% harbored multicopy $bla_{SHV-12}$. A total of 30.65% of the 124 $bla_{CTX-M-65}$-positive KPC-2-CRKP isolates carried multicopy $bla_{CTX-M-65}$ (Fig. 4A, a). In addition, multiple copies of $bla_{KPC-2}$ were found more frequently in Extended-Spectrum β-Lactamases (ESBL)-negative isolates than in ESBL-positive isolates ($P < 0.01$) (Fig. 4A, b).

### Dynamic tandem multicopy ARGs mediated by IS26

Extensive multicopy ARGs, including $bla_{KPC-2}$, $bla_{SHV-12}$, and $bla_{CTX-M-65}$, were found on the pKPC-2 plasmids (Fig. 4B). According to the assembly results of long-read Nanopore sequencing, the copy number of $bla_{KPC-2}$ ranged from 2 to 4, the copy number of $bla_{SHV-12}$ ranged from 3 to 7, and the copy number of $bla_{CTX-M-65}$ was 2 on different pKPC-2 plasmids. Furthermore, the BLASTN results of long-read sequence reads confirmed the multicopy ARG arrays and revealed that the copy number was highly dynamic on individual pKPC-2 plasmids, with 1 to 8 copies of $bla_{KPC-2}$, 1 to 10 copies of $bla_{SHV-12}$, and 1 to 4 copies of $bla_{CTX-M-65}$ identified (Fig. 4C).

Analysis of the surrounding environment suggested that the amplification of ARGs was due to IS26. Sequencing analysis showed that the translocatable units of $bla_{KPC-2}$ and $bla_{CTX-M-65}$ were all flanked by two homodromous IS26 elements, and the repeated arrays were mediated by tandem duplication of IS26 (Fig. 4B, a through c). The single copy of $bla_{SHV-12}$ was bordered by truncated and complete IS26 elements upstream and downstream, respectively (Fig. 4B, b). Then, $bla_{SHV-12}$, carried by its surrounding IS26 elements, was inserted into an IncHI1B/IncFIB(K)-type virulence plasmid and amplified at both the insertion site and the original site due to tandem duplication of IS26.

### Effect of dynamic plasmid evolution on antimicrobial resistance and virulence

Among the five strains isolated from Bed unit 16, dynamic plasmid evolution, including both homologous recombination and HGT with amplification of $bla_{SHV-12}$, occurred during transmission between the environment and the patient, forming the hybrid plasmid pKB16_E2_KPC. Regarding the antibiotic resistance profile, the three isolates harboring the hybrid pKPC-2 plasmid were all resistant to ceftazidime/avibactam (CZA), with an eightfold increase in MIC compared to the other two isolates from Bed 16. Furthermore, conjugation assays showed that pKB16_E2_KPC and pKB16_E2_SHV, both of which carried multicopy $bla_{SHV-12}$, could increase the MIC of CZA fourfold in J53 (Table S2). Regarding the virulence profile, assays using the *Galleria mellonella* larvae infection model found that the conjugant J53_pKB16_E2_KPC showed greater virulence than its ancestral isolate J53 (Fig. 5e).

The estimated copy number in the 237 ST11 KPC-2-CRKP isolates confirmed that amplification of ARGs, including $bla_{KPC-2}$, $bla_{SHV-12}$, and $bla_{CTX-M-65}$, occurred frequently (Table S1; Fig. 4A). Antimicrobial susceptibility tests (ASTs) of novel β-lactam/lactamase inhibitor combinations were performed on the 237 KPC-2-CRKP isolates to investigate the effect of ARG amplification on antimicrobial resistance. The results showed that CRKP harboring multicopy $bla_{KPC-2}$ had significantly lower susceptibility to CZA ($P < 0.01$), Meropenem/Vaborbactam (MEV) ($P < 0.01$), and Imipenem/Relebactam (IMR) ($P < 0.05$). Then, to eliminate the effects of multicopy $bla_{KPC-2}$, the relationship between multicopy ESBL genes and antimicrobial resistance was investigated among isolates with fewer than 2 $bla_{KPC-2}$ copy numbers, and the results suggested that multicopy $bla_{SHV-12}$ significantly decreased the susceptibility of CRKP to CZA ($P < 0.0001$) (Fig. 5a through c). Furthermore, the relative expression level of $bla_{SHV-12}$ in the CZA-resistant group was 6.12 times higher than that in the sensitive group ($P < 0.01$) (Fig. 5d).

## DISCUSSION

The epidemic of the plasmid-borne $bla_{KPC-2}$ gene is associated with the high-risk ST11 clone, which is the most prevalent clinical CRKP clone in China (15). In our study, 237 ST11 KPC-2-CRKP were collected in an ICU, and 32 homologous strains were subjected to

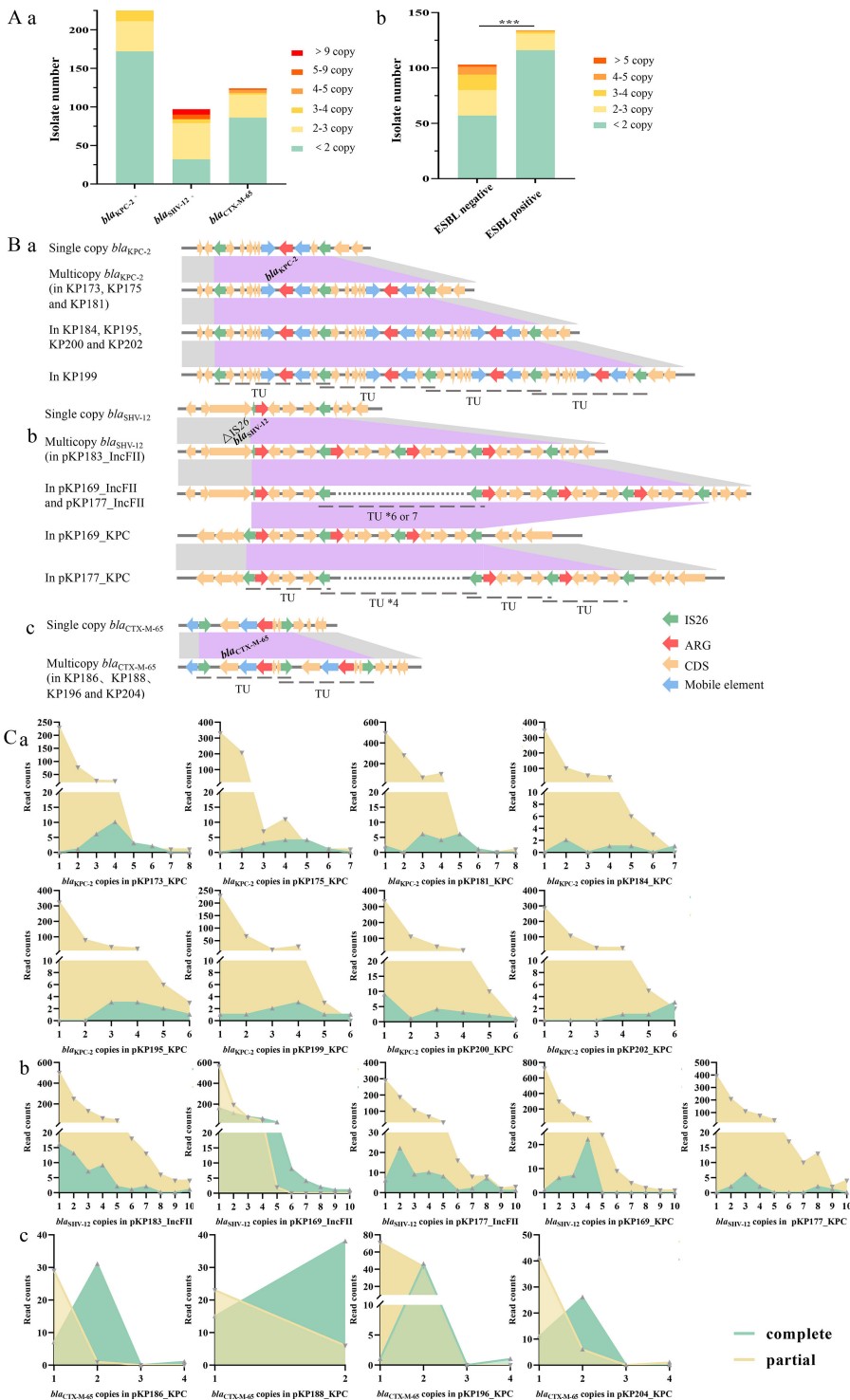

**FIG 4** Structure and confirmation of multicopy ARGs. (A, a) Distribution of strains carrying different copy numbers of ARGs, including $bla_{KPC-2}$, $bla_{SHV-12}$, and $bla_{CTX-M-65}$, among the 237 ST11 KPC-CRKP isolates. (A, b) Distribution of CRKP carrying different copy numbers of $bla_{KPC-2}$ in the ESBL-negative and ESBL-positive groups. The proportion of CRKP carrying multicopy $bla_{KPC-2}$ was significantly higher in the ESBL-negative group than in the ESBL-positive group. **$P < 0.01$. (B) Structure of tandem amplification of ARGs. Red arrows indicate ARGs. Green arrows indicate IS*26* and blue arrows indicate other IS elements. Yellow arrows indicate Coding Sequences (CDS). The translocatable units (TUs) are marked with dotted lines. (C) Confirmation and quantification of ARG copy numbers by BLASTN using long-read sequences. Limited by the length of the sequence, reads with an incomplete TU at the end were defined as partial.

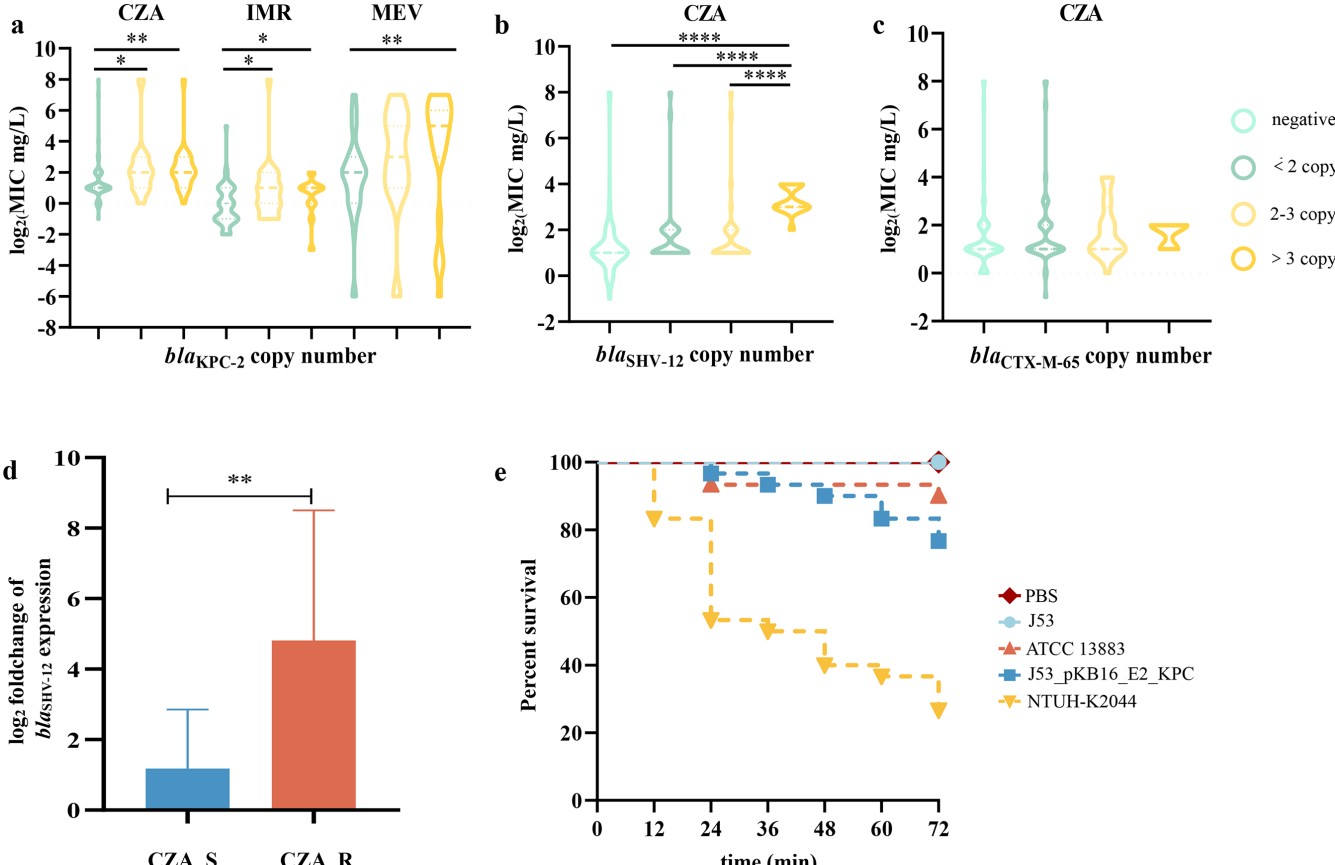

**FIG 5** The frequency of multicopy ARGs and the effect of plasmid evolution on host bacteria. (a–c) Relationship between ARG copy number and resistance to β-lactam/β-lactamase inhibitor combinations. MICs related to $bla_{KPC-2}$ copy number were analyzed in ESBL-negative isolates, and MICs related to $bla_{SHV-12}$ or $bla_{CTX-M-65}$ were analyzed in isolates carrying less than two copies of $bla_{KPC-2}$. Differences were assessed by the Mann-Whitney test using GraphPad Prism (version 7). *$P < 0.05$, **$P < 0.01$, ***$P < 0.001$, and ****$P < 0.0001$. (d) Differences in the relative expression level of $bla_{SHV-12}$ between strains in the CZA-sensitive (CZA_S) and CZA-resistant (CZA_R) groups. The experiment was repeated three times independently. The results were statistically analyzed using an unpaired $t$-test. *$P < 0.05$ and **$P < 0.01$. (e) Virulence of J53 and transconjugants in the *G. mellonella* model. The experiment was repeated in biological triplicates. Differences were assessed by log-rank (Mantel-Cox) tests.

long-read Nanopore sequencing for complete plasmid structure. Importantly, although the CRKP isolates had a close genetic relationship, their pKPC-2 plasmids evolved in a significant but trackable pathway, which involved the recombination of large resistance regions, HGT, and dynamic amplification of ARGs. Furthermore, amplifications frequently occurred in ST11 KPC-2-CRKP, which affected the isolates' susceptibility to novel β-lactamase/β-lactamase inhibitor combinations. Here, we identified the dynamic evolution processes that occurred in pKPC-2 plasmids during the clonal spread of ST11 CRKP with high resolution, and similar studies have rarely been reported.

Here, a total of 32 CRKP isolates were isolated within a span of 1 week, exhibiting a high degree of similarity with only 0 to 40 SNP differences. Due to this close similarity, distinguishing these isolates using core genomic analysis alone becomes challenging. To overcome this limitation, we employed phylogenetic analysis of the 32 ST11 CRKP strains using Roary (Fig. 1), which is well suited for clonal isolates and relies on the pan genome (16, 17). Through pangenomic epidemiology, we were able to accurately cluster the strains, with polytomies generally corresponding to differences in plasmids, such as plasmid replicon type and plasmid-associated genes. Although the CRKP isolates were closely related, the pKPC-2 plasmids were distinct and could be divided into three subgroups. The most common pKPC-2 plasmid replicon in our study was IncFII(pHN7A8)/IncR, belonging to one of the main popular pKPC-2 types

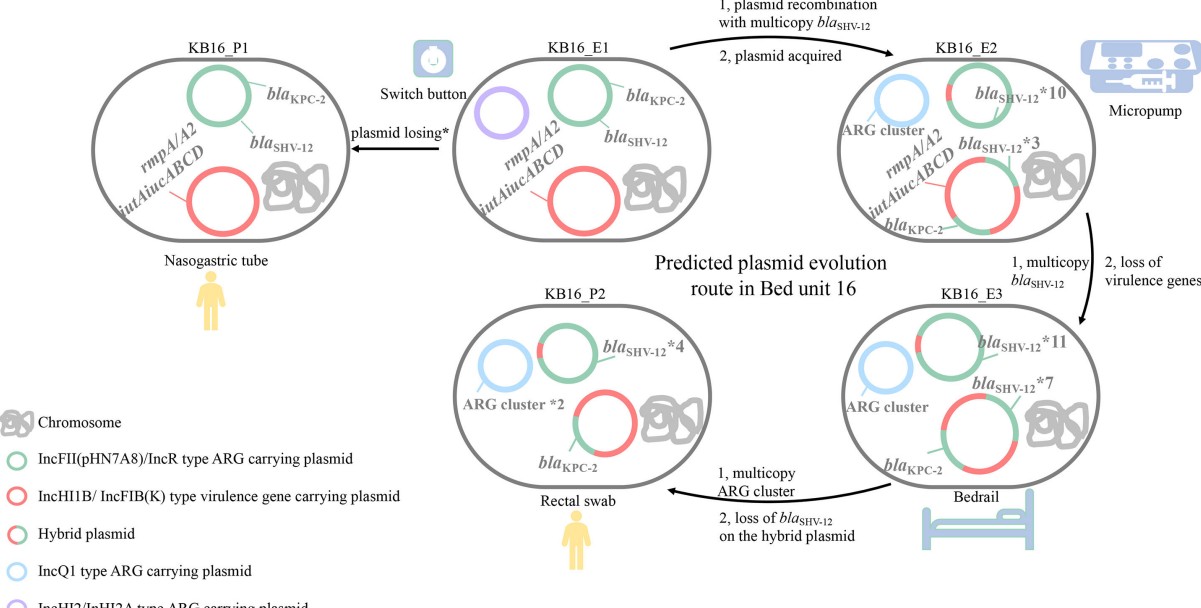

**FIG 6** Evolution of pKPC-2 plasmids in CRKP isolates from Bed 16. *, compared to the other four isolates, KB16_P1 had an inactive chromosome-borne *mgrB* (inserted at 89 bp in KB16_P1), resulting in colistin resistance. Thus, it is speculated that the evolution started from KB16_E1, not KB16_P1.

in China. In addition to IncFII(pHN7A8)/IncR, the other two dominant pKPC-2 Inc. groups in China were IncFII(pHN7A8)/Inc(pA1763-KPC) and IncFII(pKPHS2)/Inc(pA1763-KPC) (15). Notably, there were two subgroups that included pKPC-2 belonging to the IncFII(pHN7A8)/IncR type with two different resistance-encoding regions, suggesting that there could be a difference between pKPC-2 plasmids that carried the same plasmid replicons. Except for IncFII(pHN7A8)/IncR, the other replicon was the IncHIIB/IncFIB(K) type, which usually exists in virulence plasmids (18). Furthermore, it was observed that IncHIB/IncFIB(K)-type pKPC-2 was fused through recombination of IncFII(pHN7A8)/IncR-type pKPC-2 and virulence plasmids in CRKP isolates from the same bed unit, as well as HGT, which was mediated by IS*26* insertion (Fig. 3c). In addition, there was amplification of plasmid-borne ARGs *in vivo* in each subgroup, including ARGs such as *bla*$_{KPC-2}$ in Bed units 20 and 21, *bla*$_{CTX-M-65}$ in Bed unit 19, and *bla*$_{SHV-12}$ in Bed unit 16. Through the glimpse of the pKPC-2 genome, we concluded that it was rich in diversity and that it evolved rapidly, even in closely related CRKP isolates.

The detailed evolution pathways of pKPC-2 plasmids were tracked in these clonal spread CRKP isolates. In a previous study, a hybrid pKPC-2 plasmid was reported, and its formation was inferred to be associated with IS*26* based on a comparison with plasmids in GenBank (11). Conjugation assays have been reported to facilitate the formation of hybrid plasmids. Chen et al. conducted an experimental study, and plasmid recombination was observed during the conjugation process (19). Furthermore, fusion of the IncN1-F33:A-:B- plasmid and an *mcr-1*-carrying phage-like plasmid was reported with a frequency of $1.75 \times 10^{-4}$ cointegrates per transconjugant (20). Similar phenomena have been increasingly reported, indicating the important role of conjugation in the formation of hybrid plasmids, even with nonconjugative plasmids (21). For instance, Li et al. performed a conjugation assay under ceftazidime, during which they observed the fusion of a nonconjugative virulence plasmid p17-16-vir and a multidrug-resistance plasmid p17-16-CTX, resulting in the formation of a novel hybrid MDR virulence plasmid (22). However, few studies have reported the detailed evolutionary path of the clinical pKPC-2 plasmid. Here, clinical strains harboring the pKPC-2 plasmid both before and after evolution were collected, and they were found to evolve in a more diverse, complex, and dynamic way within the clinical environment (Fig. 6). Specifically,

plasmid recombination occurred during the clonal spread process; thus, fusing the novel hypervirulence and CR-resistance hybrid plasmid pKB16_E2_KPC, primarily mediated by homologous recombination, and HGT due to IS26 insertion. In contrast to the IS26-mediated insertion previously reported, this insertion led to tandem amplification of $bla_{SHV-12}$ in its original insertion sites. In previous studies, it has been demonstrated that recombination can occur between very short (<100 bp) homologous sequences (23, 24). We found that the upstream and downstream of homologous regions were 2,970 bp and 252 bp, and the upstream recombination region included transposons that belonged to the Tn3 family. In our study, the head 100 bp of the homologous segments were selected, and it turns out the coverage and identity for the left segments were 100% and 83.67%, the coverage and identity for the right segments were 99% and 86%. Despite the complexity and plasticity of pKPC-2 plasmids, their evolution can be tracked with high resolution by tracking closely related clonal spread isolates and with the aid of long-read Nanopore sequencing.

Using Nanopore sequencing, we found a wide amplification of ARGs with varying copy numbers. Schuster et al. reported that the copy numbers of the $bla_{CTX-M}$ genes varied between individual plasmids (25). We demonstrated that the dynamic copy number was associated with the flexible IS26 elements and multicopy plasmid in its host bacteria (10, 13). The amplification of ARGs, such as $bla_{KPC-2}$, $bla_{TEM}$, and $bla_{CTX-M}$, was reported sporadically, while the occurrence rate was rarely mentioned (13, 25, 26). In our study, 14 out of the 32 pKPC-2 plasmids carried multicopy ARGs. The estimated ARG copy number within the 237 ST11 KPC-2-CRKP isolates revealed that the amplification of ARGs, including $bla_{KPC-2}$, $bla_{SHV-12}$, and $bla_{CTX-M-65}$, was fairly common, and this was speculated to be a result of antibiotic selection pressure in the ICU environment. In addition, it was observed that the proportion of multicopy $bla_{KPC-2}$ was significantly higher in ESBL-negative CRKP isolates than in ESBL-positive isolates. Similarly, Sun et al. reported that the $bla_{KPC-2}$ gene copy number increased under MEV selection, except when isolates had loss-of-function mutations in both major porin genes (27). More studies are underway to clarify the influence of ESBL genes on the amplification of $bla_{KPC-2}$.

Plasmid evolution could promote the adaptation of its host bacteria, including enhancing resistance or virulence profiles (1). Here, CRKP isolates with multicopy $bla_{KPC-2}$ showed increased resistance to novel β-lactamase/β-lactamase inhibitor combinations, such as CZA, MEV, and IMR, compared with those with single-copy $bla_{KPC-2}$; this has been reported to be a result of KPC-2 overexpression (27, 28). Notably, CRKP isolates with multicopy $bla_{SHV-12}$ conferred increased resistance to ceftazidime/avibactam compared with those with a single copy, and the multicopy $bla_{SHV-12}$-encoding plasmid could increase the MIC of CZA fourfold in J53. Considering that SHV-12 has a strong ability to hydrolyze ceftazidime, the results above indicated that the increase in $bla_{SHV-12}$ gene copy number could contribute to the resistance of CZA in these clinical isolates, and this correlation has not been reported thus far, and will require further research (29). Regarding the virulence profile, the hybrid plasmid pKB16_E2_KPC carrying *iutAiucABDC* and *rmpA/A2* enhanced the virulence of J53.

Our study findings indicate that CRKP exhibited widespread colonization in both patients and environmental settings. Furthermore, we observed the evolution of carbapenem-resistance plasmids during the clonal spread of CRKP. Notably, our previously reported study revealed a correlation between long-term colonization and the risk of further infection (30). Considering these findings, we strongly recommend the implementation of regular surveillance measures for CRKP in both patient and environmental settings. Timely and systematic monitoring of CRKP prevalence and dynamics is crucial for effective infection control and prevention strategies. Additionally, it is important to prioritize timely decolonization interventions to curb the spread of CRKP. A limitation of this study is that we did not examine the novel hybrid plasmid and amplification of ARGs in the exact conditions of evolution; however, this is a fruitful

direction for further research intended to alleviate the evolution of high virulence and resistance levels in CRKP isolates.

## Conclusion

In this study, highly detailed dynamic and diverse evolution events were identified based on the genomic analysis of $bla_{KPC-2}$-harboring plasmids from clonal spread CRKP using long-read Nanopore sequencing. Here, the following three evolutionary patterns of pKPC-2 plasmids were described: (i) cointegration of virulence plasmids, (ii) horizontal gene transfer, and (iii) dynamic amplification of $bla_{KPC-2}$, $bla_{SHV-12}$, and $bla_{CTX-M-65}$. Amplification occurred with a high frequency, and multicopy $bla_{SHV-12}$ could decrease susceptibility to ceftazidime/avibactam. Notably, the formation of a novel hybrid pKPC-2 plasmid, which encoded virulence and resistance genes simultaneously, was traced and described. Taken together, these findings showed that detailed analysis of nanopore reads in homologous isolates enabled us to depict the dynamic plasmid evolution processes mediated by different MGEs. The advantageous evolution of pKPC-2 plasmids should be monitored closely to prevent their dissemination among clinical patients and environments.

## MATERIALS AND METHODS

### Bacterial isolation and antimicrobial susceptibility testing

From 1 August to 31 December 2019, a CRKP screening program was performed in the ward environments and patients in the ICU in Sir Run Run Shaw Hospital (Zhejiang Province, China), as reported before (30). Briefly, the patient's anatomic sites for sampling included rectal swab, nasointestinal tube, nasogastric tube, oral swab, tracheotomy tube, and endotracheal intubation. Environmental sampling areas included bedside areas, ward areas, and instruments in auxiliary areas. The samples were cultured on Simmion's citrate agar plate (Haibo, Qingdao, China) with 1% inositol and 2 mg/L ertapenem for initial screening of CRKP, and then a single colony of presumptive CRKP was subcultured on Mueller-Hinton agar (MHA) (Oxoid, Hampshire, UK) followed by Matrix-Assisted Laser Desorption/Ionization Time-of-Flight Mass Spectrometry (MALDI-TOF MS) (bioMerieux, France).

ASTs with ceftazidime, piperacillin/tazobactam, meropenem, imipenem, ertapenem, ceftazidime/avibactam, meropenem/vaborbactam, and imipenem/relebactam were performed according to the Clinical and Laboratory Standards Institute (CLSI) guideline M07 (31), and the results were interpreted using the breaking points of CLSI (32).

### Whole-genome sequencing and bioinformatic analyses

All CRKP isolates were sequenced using the Illumina HiSeq (Illumina, San Diego, USA) platform, and the sequence reads were assembled using Shovill 0.9.0 with the options "--trim --minlen 200 --mincov 10" (https://github.com/tseemann/shovill). The assessment of the Illumina assembly by Quast (33) revealed an average contig N50 of approximately 169 kb and an average length of the largest contig of approximately 384 kb. The quality of the assembled data was further confirmed using Kleborate. Multilocus sequence typing (MLST) was performed with mlst (https://github.com/tseemann/mlst). A minimum spanning tree of all ST11 KPC-2-CRKP isolates was generated by Ridom SeqSphere+ software version 7.2.3 (Ridom GmbH, Muenster, Germany) using the core genome MLST (cgMLST) scheme. Antibiotic resistance genes (ARGs) and plasmid replicons (rep) were identified using ABRicate with the NCBI database and the PlasmidFinder database (https://github.com/tseemann/abricate), respectively. The Kleborate databases were used for the detection of virulence factors and serotype (34). SNPs among strains were determined using Snippy version 4.4.5 (https://github.com/tseemann/snippy).

In the eighth sampling week, the most homologous ST11-KPC-2-CRKP isolates were collected, and the 32 isolates were additionally sequenced by the Oxford Nanopore MinION (Nanopore, Oxford, UK) platform to obtain complete pKPC-2 sequences. *De novo* assembly of raw nanopore reads was performed using Raven 1.8.1 (35) with error correction by polypolish (36) with the default parameters. All chromosomes and pKPC-2 plasmids were circularized completely. The phylogenetic analysis of the 32 ST11 CRKP- and $bla_{KPC-2}$-carrying plasmids was generated by Roary 3.13.0 (37) and MAFFT version 7.310 (38), respectively. The maximum-likelihood trees were constructed by IQ-TREE 2.0.3 from the alignments with the feature determined by automated detection of the best evolutionary model (39) and then visualized using the iTOL tool (40). Plasmids were annotated by Prokka and compared using blast-2.9.0+ with Easyfig version 2.2.2 for visualization (41). For the verification of duplicated ARGs on the plasmids, BLASTN was performed on all filtered long-read reads against a reference containing the tandem repeat units and relative flanking region, and the results were processed by an in-house R script (25). The short-read sequence data of 237 ST11 KPC-2-CRKP isolates were used to verify the frequency of evolution events in pKPC-2 plasmids, and the copy numbers of ARGs were estimated by CCNE (42).

## Plasmid transfer assays

Conjugation experiments of strain KB16_E2 were carried out by filter mating with sodium azide-resistant *Escherichia coli* J53 as the recipient strain (43). Overnight cultures of KB16_E2 and *E. coli* J53 were cultured in Lysogeny broth (LB) broth for 4 hours to logarithmic phase, and then their mixtures were incubated in LB broth at 37℃ for 18 hours. Potential transconjugants were selected on MHA plates containing 100 mg/L sodium azide and 8 mg/L ceftazidime for the pKB16_E2_KPC and pKB16_E2_SHV. MALDI-TOF MS (bioMerieux, France) and polymerase chain reaction (PCR) sequencing (amplifying *rep*, $bla_{SHV-12}$, $bla_{KPC-2}$, and *iucA*) of the transconjugants were subsequently performed to confirm whether the plasmid was successfully transferred to the recipient. The MIC profiles, virulence assay, and growth rates were determined for differentiation between these transconjugants and the recipient strain.

## Virulence assay

The virulence assay of J53 and its transconjugant J53_pKB16_E2_KPC was performed on a *Galleria mellonella* larvae infection model as previously described (44). ATCC 13833 was used for low virulence control, and NTUH-K2044 was used for high virulence control. Each isolate was cultured to logarithmic phase in LB broth and resuspended in phosphate-buffered saline (PBS). Ten microliters of $10^8$ CFU/mL bacterial suspension or PBS (for blank control) was injected into healthy *G. mellonella* and incubated at 37℃. Each group contained 10 *G. mellonella,* and the experiment was repeated in biological triplicates. The survival of *G. mellonella* was measured every 12 hours for 3 days. Differences in the virulence level were assessed by a log-rank (Mantel-Cox) test using GraphPad Prism (version 7). A *P* value <0.05 was considered significant.

## Real-time quantitative reverse transcription PCR

Real-time quantitative reverse transcription PCR was used to measure the expression of $bla_{SHV-12}$ in the CZA-resistant and CZA-sensitive groups. Each group contained eight $bla_{SHV-12}$-positive strains that were selected randomly among the 237 ST11 KPC-2-CRKP isolates. One strain in the susceptible group was used as the reference strain, and the *rpoB* gene was used as the internal reference (primers: qrpoB_F: TGAACAAGCTGGATT CGCCT, qrpoB_R: CGCGCAGACCAACGAATATG, qSHV12_F: AGCCGCTTGAGCAAATTAAA, qSHV12_R: GCTGGCCAGATCCATTTCTA). RNA was extracted using the PureLink RNA Mini Kit (Invitrogen, Carlsbad, CA, USA) in the exponential growth period of bacterial cells. Then, cDNA was obtained using a PrimeScript RT Reagent Kit (Takara, Kyoto, Japan). The expression level was assessed using TB Green Premix Ex Taq (Takara, Kyoto, Japan) in a

LightCycler 480 system (Roche, Rotkreuz, Switzerland) with triplicate samples for each isolate, replicating three times independently using the comparative CT method. Genes were considered differentially expressed when the |log2-fold change| was greater than 1.5 (45). The log2-fold change in gene expression was analyzed by an unpaired *t*-test on GraphPad Prism, and $P < 0.05$ was considered significant.

## ACKNOWLEDGMENTS

This work was supported by National Natural Science Foundation of China (32141001, 82272373), National Key Research and Development Program of China grant (2018YFE0102100), the Key Research Program of the Science Technology Department of Zhejiang Province (2021C03179 and 2021C03055), and Natural Science Foundation of Zhejiang Province, China (LY22H190001).

Xinhong Han and Xiaoting Hua designed the study; Xinhong Han, Junxin Zhou, and Lina Shao drafted the article; Xinhong Han and Lifei Yu obtained the data; Xinhong Han, Xiaoting Hua, and Yan Jiang analyzed and interpreted the data; Shiqi Cai, Huangdu Hu, Qiucheng Shi, Zhengan Wang, and Yunsong Yu revised the article. All authors have approved the final article.

## AUTHOR AFFILIATIONS

[1]Department of Infectious Diseases, Sir Run Run Shaw Hospital, Zhejiang University School of Medicine, Hangzhou, China
[2]Key Laboratory of Microbial Technology and Bioinformatics of Zhejiang Province, Hangzhou, China
[3]Regional Medical Center for National Institute of Respiratory Diseases, Sir Run Run Shaw Hospital, Zhejiang University School of Medicine, Hangzhou, China
[4]Department of Infectious Diseases, Affiliated Hangzhou First People's Hospital, Zhejiang University School of Medicine, Hangzhou, China

## AUTHOR ORCIDs

Xiaoting Hua https://orcid.org/0000-0001-8215-916X
Yan Jiang http://orcid.org/0000-0002-5877-9286
Yunsong Yu http://orcid.org/0000-0003-2903-918X

## FUNDING

| Funder | Grant(s) | Author(s) |
|---|---|---|
| MOST | National Natural Science Foundation of China (NSFC) | 32141001 | Yan Jiang |
| MOST | National Natural Science Foundation of China (NSFC) | 82272373 | Yan Jiang |
| MOST | National Key Research and Development Program of China (NKPs) | 2018YFE0102100 | Xiaoting Hua |
| MOST | NSFC | NSFC-Zhejiang Joint Fund | 浙江省科学技术厅 | Natural Science Foundation of Zhejiang Province (ZJNSF) | LY22H190001 | Yan Jiang |
| MOST | NSFC | NSFC-Zhejiang Joint Fund | Science and Technology Department of Zhejiang Province (浙江省科学技术厅) | 2021C03179 | Yunsong Yu |
| MOST | NSFC | NSFC-Zhejiang Joint Fund | Science and Technology Department of Zhejiang Province (浙江省科学技术厅) | 2021C03055 | Yan Jiang |

## AUTHOR CONTRIBUTIONS

Xinhong Han, Formal analysis, Methodology, Writing – original draft | Junxin Zhou, Investigation, Methodology | Lifei Yu, Supervision | Lina Shao, Methodology | Shiqi Cai, Methodology | Huangdu Hu, Writing – review and editing | Qiucheng Shi, Validation | Zhengan Wang, Writing – review and editing | Xiaoting Hua, Funding acquisition, Methodology, Writing – review and editing | Yan Jiang, Funding acquisition, Supervision, Writing – review and editing.

## DATA AVAILABILITY

All genome sequencing data pertaining to the study have been made available at the National Center for Biotechnology Information under BioProject ID PRJNA807050.

## ETHICS APPROVAL

This study was approved by ethics committee of Sir Run Run Shaw Hospital (approval number 20201217-33). This study was not considered a human research study. Therefore, no informed consent to participate was required.

## ADDITIONAL FILES

The following material is available online.

### Supplemental Material

**Supplemental figures (mSystems00924-23-S0001.pdf).** Fig. S1 and S2.
**Supplemental tables (mSystems00924-23-S0002.docx).** Tables S1 and S2.

### Open Peer Review

**PEER REVIEW HISTORY (review-history.pdf).** An accounting of the reviewer comments and feedback.

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
