## [Reviewer comments · mSystems]

Genome sequencing unveils bla_{KPC-2}-harboring plasmids as drivers of enhanced resistance and virulence in nosocomial *Klebsiella pneumoniae*

Xinhong Han, Junxin Zhou, Lifei Yu, Lina Shao, Shiqi Cai, Huangdu Hu, Qiucheng Shi, Zhengan Wang, Xiaoting Hua, Yan Jiang, and Yunsong Yu

Corresponding Author(s): Yunsong Yu, Zhejiang University School of Medicine Sir Run Run Shaw Hospital

Review Timeline:

Submission Date:	September 1, 2023
Editorial Decision:	October 9, 2023
Revision Received:	November 17, 2023
Accepted:	December 7, 2023

Editor: Sima Tokajian

Reviewer(s): Disclosure of reviewer identity is with reference to reviewer comments included in decision letter(s). The following individuals involved in review of your submission have agreed to reveal their identity: Santiago Castillo-Ramírez (Reviewer #1); Victor Gonzalez (Reviewer #2)

Transaction Report:

DOI: <https://doi.org/10.1128/msystems.00924-23>

October 9, 2023

Prof. Yunsong Yu
Zhejiang University School of Medicine Sir Run Run Shaw Hospital
Department of Infectious Diseases
3#, Qingchun east road
Hangzhou, Zhejiang 310016
China

Re: mSystems00924-23 (A genomic glimpse of *bla*_{KPC-2}-harboring plasmids revealed the dynamic evolution enhanced the resistance and virulence profiles of nosocomial *Klebsiella pneumoniae*)

Dear Prof. Yunsong Yu:

Thank you for submitting your manuscript to mSystems. We have completed our review and I am pleased to inform you that, in principle, we expect to accept it for publication in mSystems. However, acceptance will not be final until you have adequately addressed the reviewer comments.

Please add a statement regarding informed consent under ethical approval in the revised version.

Preparing Revision Guidelines

Please return the manuscript within 60 days; if you cannot complete the modification within this time period, please contact me. If you do not wish to modify the manuscript and prefer to submit it to another journal, please notify me of your decision immediately so that the manuscript may be formally withdrawn from consideration by mSystems.

Sincerely,

Sima Tokajian

Editor, mSystems

Journals Department
Reviewer comments:

Reviewer #1 (Comments for the Author):

Summary

The authors sequence, using short-read, 237 ST11 KPC-2-CRKP isolates (sampled between August, 1st, and December, 31st, 2019) and 32 CRKP isolates were also selected for long-read sequencing to analyse the transmission dynamics of pKPC-2s. Also, AST, conjugation and virulence assays were conducted to further analyse the pKPC-2s. This is an interesting study adding to the role of plasmids in the dissemination of antimicrobial drug resistance genes. However, there are some technical details that the authors want to address to enhance their paper.

Major comments

1. The authors want to provide some assembly statistics (# contigs, N50, coverage, etc.) so that the reader can appreciate the quality of the data.
2. To be completely sure about the proper species assignment please conduct an average nucleotide identity analysis considering at least one type strain of *K. pneumoniae*.
3. Phylogeny, in Figure 1, has several polytomies within both clades. Recently, it has been suggested that pan-genomic epidemiology (the use of both core and accessory genomes) can be used to resolve this situation. See refs below

<https://pubmed.ncbi.nlm.nih.gov/35544058/>

<https://pubmed.ncbi.nlm.nih.gov/34282943/>

4. How recombination was inferred? The authors want to use a program to properly detect recombination in the sequences analysed.
5. It's not clear to me how phylogeny in Figure 2 was constructed, what this based on the relaxase or based on some Mash distance?

Minor comments

Modify the title for clarity. I would suggest something like ...

"Genome sequencing unveils blaKPC-2 plasmids as drivers of enhanced resistance and virulence in nosocomial *Klebsiella pneumoniae*"

Line 43: should be "Long-read sequencing was"

Line 64: should be just "*K. pneumoniae*"

Line 79: should be "Using long-read sequencing"

Lines 218: what do you mean by recombination traces? Did you use any specific program to detect recombination?

Regarding Table S2 please highlight resistant cases in bold.

Reviewer #2 (Comments for the Author):

The manuscript "Msystems00924-23" by Xinhong Han and co-authors delineates the evolutionary trajectory of blaKPC plasmids in clinical isolates of the pathogenic bacteria *K. pneumoniae*. Their combined approach, involving microbial isolation from both patients and ward environments, along with the utilization of Illumina and Nanopore sequencing, as well as antibiotic and virulence testing, facilitated the identification of blaKPC plasmids with distinctive structural configurations. Furthermore, the authors conducted an exhaustive sequence analysis to deduce the potential recombination mechanisms underlying the dissemination of blaKPC-2 in different plasmids.

There are limited reports focusing on tracking the evolution of antibiotic resistance plasmids in local nosocomial contexts. Additionally, it is known that assembling plasmids encoding antibiotic resistance with short sequencing reads poses a challenge. The authors effectively resolve this limitation by employing a Nanopore sequencing strategy. The study is well-executed and well-articulated. Nevertheless, some details in the text and the explanation of results could enhance its presentation.

1. I understand that the studied *K. pneumoniae* strains were based on the dissemination of a single clonal lineage that originated from the highest peak of sampling. The evidence presented highlights the close relationships among these isolates. Did the authors investigate the genetic or phenotypic relationships (antibiotic resistance, plasmids) of these isolates with the rest of the isolates obtained during the screening period? Is the propensity to vary a property of the clone studied, or is there a possibility of parallel evolution in other isolates?
2. Since CRKP colonies were obtained at the infection sites or in the environmental ward, where did the recombinant plasmids originate? Within the patient or in any other place?
3. In section 2.2, the information on Illumina and Nanopore sequencing should be complemented by adding the overall quality, sequence read length, and coverage of the plasmid assemblies. It is important to report them for a correct evaluation of the recombinant junctions and gene duplications proposed in the models.
4. L. 217. What exactly does "recombination traces" mean?
5. In the section about the formation of the hybrid pKPC-2 plasmid (l. 225-246), homologous recombination via identical sequences shared by the plasmids is the normal way to explain the results. However, the identity of the sequences is slightly lower than optimal for high-frequency recombination. Then the event may occur at low frequencies or the intervention of some recombinase from a transposon is needed. Would you comment on this point?
6. The emergence of plasmid variants in a short period of time suggests that antibiotic selective pressures may justify their appearance at high frequencies. Are there clinical data on the antibiotic therapy applied during the sampling period that could have provoked the observed pattern?
7. While recommendations for surveillance of antibiotic-resistant strains could be drawn from this work, these are not explicitly commented on in the text.
8. l. 369. What do you mean with "exact conditions if evolution"?

Reviewer #3 (Comments for the Author):

The current widespread blaKPC-2-bearing plasmids in *Klebsiella pneumoniae* constitute a great public concern. Understanding the evolution pathways of such plasmids in clinical settings is of importance to figure out the control methods. Also, the current convergence of MDR plasmids and virulence plasmids in KP is another severe threat. This study utilized the genomic analysis including long-read sequencing data analysis to decipher the dynamic evolution pathways under the ICU ward, shedding insight on the rapid transmission of such MDR plasmids. The study is a comprehensive work covering sampling, AST, genome sequencing analysis, gene expression assay, conjugation assay and virulence detection. Several minor suggestions were provided here for reference.

1. The figures were not in a right direction.
2. What's the reason of picking up the isoates for long-read sequencing?
3. For the resistance gene tandem repeats, are they stable in copy number or not for a single strain?
4. Conjugation assay has been reported to facilitate the formation of hybrid plasmids. Authors could cite more refernces to highlight the wide existence of such phenomenon. More discussion in this area is suggested.
5. When the resistance genes are mutiply amplified in the evolved strains, how about the MICs of corresponding antimicrobials?
6. pKPC-2 plasmid seems not to be a specific plasmid, but including all plasmids harboring blaKPC-2. Authors should check the consistence of such presentation.

Dear Editor and Reviewers:

Thank you for taking our manuscript (mSystems00924-23) into consideration. We appreciate your comments and suggestions which are valuable for improving our paper and helpful in promoting the importance of our work.

According to the comments and suggestions, a word-by-word revision has been made on the manuscript and amendments are highlighted by using the highlighter tool in the revised version. We make a point-by-point response to the comments below and hope it is acceptable for publication in “*mSystems*”.

Reviewer #1 (Comments for the Author):

Summary

The authors sequence, using short-read, 237 ST11 KPC-2-CRKP isolates (sampled between August, 1st, and December, 31st, 2019) and 32 CRKP isolates were also selected for long-read sequencing to analyse the transmission dynamics of pKPC-2s. Also, AST, conjugation and virulence assays were conducted to further analyse the pKPC-2s. This is an interesting study adding to the role of plasmids in the dissemination of antimicrobial drug resistance genes. However, there are some technical details that the authors want to address to enhance their paper.

Major comments

1. The authors want to provide some assembly statistics (# contigs, N50, coverage, etc.) so that the reader can appreciate the quality of the data.

****Thank you for your valuable suggestion. We have added the assembly statistics of Illumina sequencing and nanopore sequencing in the Methods section in lines 348-352 and lines 363-364, respectively. For Illumina sequencing data, the sequence reads were *de novo* assembled using Shovill 0.9.0 with the options ‘--trim --minlen 200 --mincov 10’. The assessment of the Illumina assembly revealed an average contig N50 of approximately 169 kb and an average length of the largest contig of approximately 384 kb. The quality of the assembled data was further confirmed using Kleborate.**

Regarding the Nanopore sequencing data, the raw reads were *de novo* assembled using Raven

1.8.1, followed by error correction using Polypolish with default parameters. All chromosomes and pKPC-2 plasmids of the 32 strains were circularized completely.

The quality assessment of the assembled data was performed by Quast, and the results are listed in Supplementary material for review_1.

2. To be completely sure about the proper species assignment please conduct an average nucleotide identity analysis considering at least one type strain of *K. pneumoniae*.

**Thank you for your suggestion. We performed average nucleotide identity (ANI) analysis on the 237 isolates using FastANI, with *K. pneumoniae* ATCC 13883 as the reference strain. The ANI values of the 237 isolates ranged from 98.82 to 99.14 when compared to ATCC 13883. The results of FastANI analysis are listed in Supplementary material for review_1. Additionally, we employed Kleborate for species identification. The results confirmed that all 237 strains belong to *K. pneumoniae* sensu stricto, without any subspecies identified.

3. Phylogeny, in Figure 1, has several polytomies within both clades. Recently, it has been suggested that pan-genomic epidemiology (the use of both core and accessory genomes) can be used to resolve this situation. See refs below (<https://pubmed.ncbi.nlm.nih.gov/35544058/>; <https://pubmed.ncbi.nlm.nih.gov/34282943/>)

**We appreciate your valuable comments and agree with your suggestion. Here, a total of 32 CRKP isolates were isolated within a span of one week, exhibiting a high degree of similarity with only 0 to 40 SNP differences. Due to this close similarity, distinguishing these isolates using core genomic analysis alone becomes challenging. To overcome this limitation, we employed phylogenetic analysis of the 32 ST11 CRKP strains using Roary (Figure 1), which is well suited for clonal isolates and relies on the pan genome. Through pangenomic epidemiology, we were able to accurately cluster the strains, with polytomies generally corresponding to differences in plasmids, such as plasmid replicon type and plasmid-associated genes. Our primary objective was to trace the evolutionary pathway of the high-resolution pKPC-2 plasmid. Since the two clusters depicted in Figure 1 initially enable differentiation of strains carrying pKPC-2, we did not further subdivide the subclades. The aforementioned information was further elaborated upon in lines 227-234, with reference to the two cited sources.

4. How recombination was inferred? The authors want to use a program to properly detect recombination in the sequences analysed.

**Thank you for your comment. In our study, we inferred the recombination of pKPC-2 plasmids through manual sequence analysis, incorporating modifications from previously described methods (1, 2). We employed several bioinformatics tools, namely, Prokka, Easyfig, and CLC Genomics Workbench, during the analysis process.

First, the complete sequences of pKPC-2 were obtained by combining Illumina short-read sequencing and nanopore long-read sequencing. These sequences were then annotated using Prokka, and manual corrections were made as necessary. Subsequently, we compared the annotated plasmids using blast-2.9.0+ and visualized the results with Easyfig v2.2.2. This visualization allowed us to identify potential recombination segments between the plasmids. To further investigate the recombination events, we employed CLC Genomics Workbench for comparative analysis, which provided DNA nucleotide sequences. Within these sequences, we specifically searched for evidence of recombination around the identified recombination segments, including insertion target sites and homologous recombination fragments.

By following this approach, we inferred recombination events in the pKPC-2 plasmids without utilizing dedicated recombination detection software. In this study, there were markers that supported the manual analysis results, including the 8-bp target site of IS26, site-specific recombination markers and Tn3-related recombination.

5. It's not clear to me how phylogeny in Figure 2 was constructed, what this based on the relaxase or based on some Mash distance?

**Thank you for your review. The phylogeny in Figure 2 was constructed using the complete sequence of pKPC-2 plasmids. The alignment of the sequences was performed using MAFFT v7.310 with the maximum-likelihood method. The best-fit analysis model for the data was determined automatically based on the Bayesian Information Criterion (BIC), and the selected model was TPM2+F+G4. Once the sequence alignment was obtained, the phylogenetic tree was constructed using IQ-TREE 2.0.3. Based on this model and the aligned sequences, IQ-TREE 2.0.3 was used to construct the phylogenetic tree using the maximum likelihood method.

Minor comments

Modify the title for clarity. I would suggest something like ...

"Genome sequencing unveils blaKPC-2 plasmids as drivers of enhanced resistance and virulence in nosocomial *Klebsiella pneumoniae*" 
**Thank you for your valuable suggestion. The title has been modified according to your suggestion.

Line 43: should be "Long-read sequencing was"

**The expression has been modified.

Line 64: should be just "K. pneumoniae"

**The word has been modified.

Line 79: should be "Using long-read sequencing"

**The expression has been modified.

Lines 218: what do you mean by recombination traces? Did you use any specific program to detect recombination?

**Thank you for your review. 1. The sentence has been modified to “The two subgroups of IncFII/IncR type plasmids exhibited similar skeletons, with evidence of recombination events”. Subgroup 2 contained two additional segments compared to the common segments found in both subgroups. Around these additional segments, a 28-bp marker of site-specific recombination was identified, as reported previously (3). 2. In the study, the recombination analysis was conducted manually following a previously established method for performing comparative analysis to identify recombination events. Bioinformatics tools were used during the analysis process, including Prokka, Easyfig and CLC Genomics Workbench.

Regarding Table S2 please highlight resistant cases in bold.

**Thank you for your suggestion. The resistant isolates are highlighted in bold.

Reviewer #2 (Comments for the Author):

The manuscript "Msystems00924-23" by Xinhong Han and co-authors delineates the evolutionary trajectory of *bla*_{KPC-2} plasmids in clinical isolates of the pathogenic bacteria *K. pneumoniae*. Their combined approach, involving microbial isolation from both patients and ward environments, along with the utilization of Illumina and Nanopore sequencing, as well as antibiotic and virulence testing, facilitated the identification of *bla*_{KPC-2} plasmids with distinctive structural configurations.

Furthermore, the authors conducted an exhaustive sequence analysis to deduce the potential recombination mechanisms underlying the dissemination of *bla*_{KPC-2} in different plasmids.

There are limited reports focusing on tracking the evolution of antibiotic resistance plasmids in local nosocomial contexts. Additionally, it is known that assembling plasmids encoding antibiotic resistance with short sequencing reads poses a challenge. The authors effectively resolve this limitation by employing a Nanopore sequencing strategy.

The study is well-executed and well-articulated. Nevertheless, some details in the text and the explanation of results could enhance its presentation.

1. I understand that the studied *K. pneumoniae* strains were based on the dissemination of a single clonal lineage that originated from the highest peak of sampling. The evidence presented highlights the close relationships among these isolates. Did the authors investigate the genetic or phenotypic relationships (antibiotic resistance, plasmids) of these isolates with the rest of the isolates obtained during the screening period? Is the propensity to vary a property of the clone studied, or is there a possibility of parallel evolution in other isolates? ?

****Thank you for your comments. This is a very intriguing question and needs further investigation based on extended application of long-read nanopore sequencing. Here, we initially explored this subject using the results obtained from short-read Illumina sequencing. During the screening period, a total of 420 CRKP were isolated, among which the predominant epidemic clones were ST11 KPC-2-producing CRKP (KPC-CRKP) (237/420) and ST15 OXA-232-producing CRKP (OXA-CRKP) (106/420), as illustrated in the Figure R. CRKP strains belonging to other ST types were sporadically isolated, including 30 ST15 KPC-CRKP strains, 16 ST1 strains, 15 ST133 strains, 12 ST556 strains, 2 ST268 strains and 2 non-carbapenemase-producing ST11 strains. Therefore, we chose to focus on ST15 OXA-232-CRKP for comparative analysis.**

Similar to ST11 KPC-CRKP, ST15 OXA-CRKP strains exhibited clonal spread with a close genetic relationship (with 0 to 45 SNP differences). However, the two clones differed in terms of antibiotic resistance profiles, plasmid types and the evolutionary characteristics of carbapenem resistance plasmids. Regarding the antibiotic resistance profile, the two clones displayed discrepancies in their susceptibility to meropenem, imipenem, ceftazidime/avibactam, amikacin, meropenem/vaborbactam, imipenem/relebactam and fosfomycin (Table R). ST11 KPC-CRKP

contained more plasmid replicon types than ST15 OXA-CRKP. Notably, multiple evolution events were observed in the carbapenem-resistance plasmid pKPC-2 within the ST11 KPC-CRKP, including ARG duplications and homologous recombinations. In contrast, the carbapenem resistance plasmid pOXA-232 in ST15 OXA-CRKP remained stable, with a size of 6.14 kb, and no evolution events were observed in the small plasmid, as reported in our previously published study (4).

Figure R. Molecular epidemiological characteristics of 420 strains of CRKP isolated during the screening period

Table R. Comparison of antimicrobial susceptibility between ST15 OXA-CRKP and ST11 KPC-CRKP

Antibiotics*	ST15 OXA-CRKP (%)			ST11 KPC-CRKP (%)		
	S	I	R	S	I	R
FOX	0	0	100	0.84	0	99.16
CAZ	0	3.78	96.22	0	0	100
FEP	0	1.89	98.11	0	2.11	97.89
PTZ	0	0	100	0.42	0	99.58

F/S	0	0	100	0.42	0.42	99.16
MEM	0.94	8.49	90.57	0	0.84	99.16
IPM	9.44	33.96	56.60	0	0.84	99.16
ETP	0	0	100	0	0	100
AZT	0	0	100	0.84	0	99.16
AMI	0	0	100	59.92	0	40.08
LEV	0	0	100	0	0.42	99.58
CIP	0	0	100	0	0	100
C/T	0	0	100	0	0	100
CZA	100	-	0	91.98	-	8.02
I/R	8.50	41.50	50	65.04	24.47	10.13
MEV	9.43	12.26	78.30	63.29	15.19	21.52
FOS	83.02	9.43	7.55	32.91	47.68	19.41
COL	-	93.40	6.60	-	89.45	10.55
TGC	82.05	9.40	8.55	87.34	9.28	3.38

2. Since CRKP colonies were obtained at the infection sites or in the environmental ward, where did the recombinant plasmids originate? Within the patient or in any other place?

****Thank you for your comments. This is a very valuable question and needs further research. In this study, we isolated 5 CRKP strains from both patients and environmental settings in bed Unit 16. Among these strains, three were found to carry recombinant plasmids. Specifically, we isolated the recombinant pKB16_E2_KPC plasmid from a micropump, which was employed for microinjection drug delivery. However, the exact origin of the plasmid could not be determined through our observational study. The presence of CRKP isolates in both patients and environmental settings suggests the possibility of a recombination process occurring either within the environment or within patients and subsequently spreading to the micropump. Previous studies have reported the isolation of recombination plasmids both from patients and through in vitro conjugation assays (5, 6). Although conjugation has been shown to facilitate the formation of recombinant plasmids, the specific location of their origin has not been reported and requires further research (7, 8). Based on our observations and plasmid analysis, we have described the putative recombination process in Figure 6.**

3. In section 2.2, the information on Illumina and Nanopore sequencing should be complemented by adding the overall quality, sequence read length, and coverage of the

plasmid assemblies. It is important to report them for a correct evaluation of the recombinant junctions and gene duplications proposed in the models.

**Thank you for your valuable comments. We have added the assembly statistics of Illumina sequencing and nanopore sequencing in the Methods section in lines 348-352 and lines 363-364, respectively. For Illumina sequencing data, the sequence reads were *de novo* assembled using Shovill 0.9.0 with the options '--trim --minlen 200 --mincov 10'. The assessment of the Illumina assembly revealed an average contig N50 of approximately 169 kb and an average length of the largest contig of approximately 384 kb. The quality of the assembled data was further confirmed using Kleborate.

Regarding the Nanopore sequencing data, the raw reads were *de novo* assembled using Raven 1.8.1, followed by error correction using Polypolish with default parameters. The plasmids were obtained by *de novo* assembly of nanopore sequencing, and all chromosomes and pKPC-2 plasmids of the 32 strains were circularized completely.

The quality assessment of the assembled data was performed by Quast and the results are listed in Supplementary material for review_1.

4. L. 217. What exactly does "recombination traces" mean?

**Thank you for your review. I apologize for the confusion caused by my previous expression. Recombination traces refer to the various elements involved in the recombination process, such as recombination fragments, recombination sites, and recombination markers. A specific example is the recombination process observed in pKB16_E2_KPC, which involves the fusion of IncFII(pHN7A8)/IncR-type pKPC-2 and virulence plasmids in CRKP isolates. This fusion is mediated by the IS26 insertion of *bla*_{SHV-12} fragments and Tn3-related homologous recombination of *bla*_{KPC-2} fragments.

5. In the section about the formation of the hybrid pKPC-2 plasmid (l. 225-246), homologous recombination via identical sequences shared by the plasmids is the normal way to explain the results. However, the identity of the sequences is slightly lower than optimal for high-frequency recombination. Then the event may occur at low frequencies or the intervention of some recombinase from a transposon is needed. Would you comment on this point?

**Thank you for your comments. According to a study by Lu *et al.* , homology recombination can

occur between short homology sequences (> 20 bp) with the assistance of widely distributed bacterial strand exchange proteins, such as RecA (9). Then, we focused on the head 100 bp homology segments and found that the left segments had 100% coverage and 84% identity, while the right segments had 99% coverage and 86% identity. Similarly, Zhao et al. reported plasmid recombination between a pLVPK-like virulence plasmid and a conjugative blaKPC-2-carrying plasmid, which involved a 43 bp homology segment with 84% identity (10). This suggests that an identity of over 80% may be efficient for homologous recombination.

Furthermore, several studies have observed the involvement of recombinase enzymes from transposons in plasmid formation (11 – 13). Additionally, Li *et al.* reported plasmid recombination induced by intron reverse transcriptase and the mobile element ISS*hes11* (7). In our study, the homologous segments were located in Tn*As1* and Tn*As2*, both of which belong to the Tn3 family transposons. It is possible that the recombination observed in our research involves the intervention of a site-specific recombinase called resolvase, which is derived from the Tn3 transposon.

6. The emergence of plasmid variants in a short period of time suggests that antibiotic selective pressures may justify their appearance at high frequencies. Are there clinical data on the antibiotic therapy applied during the sampling period that could have provoked the observed pattern?

**During the screening period, a total of 39 patients were colonized by KPC-2-producing carbapenem-resistant *Klebsiella pneumoniae* (CRKP). To investigate the antibiotic factors contributing to the emergence of plasmid variants, we reviewed the medical records of these patients to identify the antibiotics prescribed prior to the isolation date. Among the 39 patients, the majority (29/39) had received treatment with piperacillin/tazobactam (TZP) and/or cefoperazone/sulbactam, while 21/39 patients had been treated with carbapenems (Supplementary material for review_2). Furthermore, we took a patient in B16 as an example, where CRKP strains carry multiple plasmid variants. A month before the isolation of the plasmid variants, this particular patient had undergone a 9-day course of TZP (4.5 g q8h), followed by a 13-day course of TZP (4.5 g q8h) and a 3-day course of biapenem (0.3 g q12h). Furthermore, TZP was still being used on the isolation day. Notably, in a separate study, TZP-resistant *E. coli* was isolated after a 19-day treatment with TZP, accompanied by amplification of the blaTEM-1B gene (14).

Additionally, an experimental evolution assay demonstrated the amplification of *bla*_{KPC-2} under the selective pressures of β -lactam antibiotics, including ceftazidime, meropenem, or moxalactam. (15). Based on these observations, we speculated that the overuse of β -lactam/lactamase inhibitor combinations and carbapenem antibiotics was associated with the evolution of pKPC-2. These findings highlight the potential role of antibiotic selective pressures in driving the emergence of pKPC-2 plasmid variants. However, importantly, these conclusions are based on observational data, and further research is warranted to establish a more comprehensive understanding of the relationship between antibiotic usage and the evolution of pKPC-2.

7. While recommendations for surveillance of antibiotic-resistant strains could be drawn from this work, these are not explicitly commented on in the text.

**Thank you for your suggestion. Our study findings indicate that CRKP exhibited widespread colonization in both patients and environmental settings. Furthermore, we observed the evolution of carbapenem-resistance plasmids during the clonal spread of CRKP. Notably, our previously reported study revealed a correlation between long-term colonization and the risk of further infection (4). Considering these findings, we strongly recommend the implementation of regular surveillance measures for CRKP in both patient and environmental settings. Timely and systematic monitoring of CRKP prevalence and dynamics is crucial for effective infection control and prevention strategies. Additionally, it is important to prioritize timely decolonization interventions to curb the spread of CRKP. The recommendations for surveillance are supplied in lines 304-312.

8.1. 369. What do you mean with "exact conditions if evolution"?

**Thank you for your comments. I apologize for the confusion of the expression. When I mentioned "exact conditions of evolution", I specifically addressed the circumstances and environmental factors that contribute to the process of pKPC-2 evolution in bacteria. These conditions include the characteristics of the host bacteria, the structure of the pKPC-2 plasmid, and exposure to antibiotics. In our study, we observed pKPC-2 evolution in ST11 CRKP. Moving forward, we plan to investigate the evolution characteristics of pKPC-2 in other ST types and explore the potential impact of mobile elements on this process. It has been reported that antibiotic usage facilitated the evolution of CRKP. However, the specific details regarding the type of antibiotic, the duration of antibiotic usage, and the concentration of the antibiotic that induces the

evolution of pKPC-2 have not been confirmed. To gain a more comprehensive understanding of the conditions that promote pKPC-2 plasmid evolution, further *in vitro* experiments are needed.

Reviewer #3 (Comments for the Author):

The current widespread blaKPC-2-bearing plasmids in *K. pneumoniae* constitute a great public concern. Understanding the evolution pathways of such plasmids in clinical settings is of importance to figure out the control methods. Also, the current convergence of MDR plasmids and virulence plasmids in KP is another severe threat. This study utilized the genomic analysis including long-read sequencing data analysis to decipher the dynamic evolution pathways under the ICU ward, shading insight on the rapid transmission of such MDR plasmids. The study is a comprehensive work covering sampling, AST, genome sequencing analysis, gene expression assay, conjugation assay and virulence detection. Several minor suggestions were provided here for reference.

1. The figures were not in a right direction.

**Thank you for your comments. Due to an error during the merging process, the direction of the figures became mixed. We have now rectified this mistake and made the necessary corrections.

2. What's the reason of picking up the isoates for long-read sequencng?

**Thank you for your comments. Our objective was to provide a comprehensive description of the population structure and evolutionary characteristics of pKPC-2 in clinical settings. However, the use of short-read whole-genome sequencing (WGS) poses challenges in achieving complete and accurate reconstruction of plasmid structures. To overcome this limitation, we proposed the utilization of nanopore sequencing to obtain full-length plasmid sequences for a more thorough analysis of plasmid evolution. Furthermore, the widespread distribution and rapid changes in mobile genetic elements (MGEs) make it difficult to discern the specific evolutionary processes associated with pKPC-2, given its complex genetic structure. In light of this, we specifically selected genetically indistinguishable CRKP strains isolated from an ICU in the highest peak of sampling day. Along with usage of long-read sequencing, the evolution of the pKPC-2 plasmids could be tracked in a distinguishable way.

3. For the resistance gene tandem repeats, are they stable in copy number or not for a single

strain?

**Thank you for your comments. We conducted BLASTN analysis on the filtered long-read reads, comparing them to a reference containing the tandem repeat units and their respective flanking regions. The results revealed that the copy number of resistance gene tandem repeats exhibited instability within a single strain, as depicted in Figure 4C. For instance, in the case of *bla*_{KPC-2} in pKP173_KPC, the copy number varied from 1 to 8, with the majority of individual plasmids carrying 4 copies. Similarly, Schuster *et al.* reported that the copy numbers of the *bla*_{CTX-M} genes ranged from 1 to 5 copies between individual plasmids in a single multidrug-resistant *Escherichia coli* isolate (16).

4. Conjugation assay has been reported to facilitate the formation of hybrid plasmids. Authors could cite more referneces to highlight the wide existence of such phenomenona. More discussion in this area is suggested.

**Thank you for your valuable suggestion. The relevant discussion has been added in lines 251-263. “In a previous study, a hybrid pKPC-2 plasmid was reported, and its formation was inferred to be associated with IS26 based on a comparison with plasmids in GenBank (5). Conjugation assays have been reported to facilitate the formation of hybrid plasmids. Chen *et al.* conducted an experimental study, and plasmid recombination was observed during the conjugation process (6). Furthermore, fusion of the IncN1-F33:A-:B- plasmid and an *mcr-1*-carrying phage-like plasmid was reported with a frequency of 1.75×10^{-4} cointegrates per transconjugant (1). Similar phenomena have been increasingly reported, indicating the important role of conjugation in the formation of hybrid plasmids, even with nonconjugative plasmids (8). For instance, Li *et al.* performed a conjugation assay under ceftazidime, during which they observed the fusion of a nonconjugative virulence plasmid p17-16-vir and a multidrug-resistance plasmid p17-16-CTX, resulting in the formation of a novel hybrid MDR virulence plasmid (7).”

5. When the resistance genes are mutiply amplified in the evolved strains, how about the MICs of corresponding antimicrobials?

**Thank you for your comments. The amplification of *bla*_{KPC-2} and *bla*_{SHV-12} was demonstrated to decrease the susceptibility of CRKP to certain antibiotics, including ceftazidime/avibactam (CZA), meropenem/varbobactam (MEV) and imipenem/relebactam (IMR). Specifically, conjugation assays have demonstrated that pKB16_E2_KPC and pKB16_E2_SHV, both of which carry

multicopy *bla*_{SHV-12}, could increase the MIC of CZA by 4-fold in J53. Furthermore, antimicrobial susceptibility testing (AST) of novel β -lactam/lactamase inhibitor combinations was performed on the 237 KPC-2-CRKP isolates. The results revealed that CRKP strains harboring multiple copies of *bla*_{KPC-2} exhibited significantly reduced susceptibility to CZA ($p < 0.01$), MEV ($p < 0.01$) and IMR ($p < 0.05$). Then, to eliminate the effects of multicopy *bla*_{KPC-2}, the relationship between multicopy ESBL genes and antimicrobial resistance was investigated among isolates with fewer than 2 *bla*_{KPC-2} copy numbers, and the results suggested that multicopy *bla*_{SHV-12} significantly decreased the susceptibility of CRKP to CZA ($p < 0.0001$) (Fig. 5abc).

6. pKPC-2 plasmid seems not to be a specific plasmid, but including all plasmids harboring *bla*_{KPC-2}. Authors should check the consistence of such presentation.

**Thank you for your suggestion. Yes, the pKPC-2 plasmid is an abbreviated representation of the *bla*_{KPC-2}-harboring plasmid in this study, encompassing all plasmids that carry the *bla*_{KPC-2} gene. We have verified the accuracy of this presentation. Furthermore, we have made modifications to the expressions in Lines 123, 147, 195, 302, 320, and 596 as suggested.

References:

1. He D, Zhu Y, Li R, Pan Y, Liu J, Yuan L, Hu G. 2019. Emergence of a hybrid plasmid derived from IncN1-F33:A–:B– and *mcr-1*-bearing plasmids mediated by IS26. *Journal of Antimicrobial Chemotherapy* 74:3184–3189.
2. Li R, Cheng J, Dong H, Li L, Liu W, Zhang C, Feng X, Qin S. 2020. Emergence of a novel conjugative hybrid virulence multidrug-resistant plasmid in extensively drug-resistant *Klebsiella pneumoniae* ST15. *International Journal of Antimicrobial Agents* 55:105952.
3. Xu Y, Zhang J, Wang M, Liu M, Liu G, Qu H, Liu J, Deng Z, Sun J, Ou H-Y, Qu J. 2021. Mobilization of the nonconjugative virulence plasmid from hypervirulent *Klebsiella pneumoniae*. *Genome Medicine* 13:119.
4. Han X, Chen Y, Zhou J, Shi Q, Jiang Y, Wu X, Quan J, Hu H, Wang Q, Yu Y, Fu Y. 2022. Epidemiological Characteristics of OXA-232-Producing Carbapenem-Resistant *Klebsiella pneumoniae* Strains Isolated during Nosocomial Clonal Spread Associated with Environmental Colonization. *Microbiology Spectrum* 1–13.
5. Jin L, Wang R, Gao H, Wang Q, Wang H. 2021. Identification of a Novel Hybrid Plasmid

- Encoding KPC-2 and Virulence Factors in *Klebsiella pneumoniae* Sequence Type 11. *Antimicrobial Agents and Chemotherapy* 65:1–6.
6. Chen K, Xie M, Chan EW-C, Chen S. 2022. Delineation of ISEcp1 and IS26-Mediated Plasmid Fusion Processes by MinION Single-Molecule Long-Read Sequencing. *Frontiers in Microbiology* 12.
 7. Li R, Cheng J, Dong H, Li L, Liu W, Zhang C, Feng X, Qin S. 2020. Emergence of a novel conjugative hybrid virulence multidrug-resistant plasmid in extensively drug-resistant *Klebsiella pneumoniae* ST15. *International Journal of Antimicrobial Agents* 55:105952.
 8. Shan X, Yang M, Wang N, Schwarz S, Li D, Du X-D. 2022. Plasmid Fusion and Recombination Events That Occurred during Conjugation of *poxxA*-Carrying Plasmids in Enterococci. *Microbiology Spectrum* 10:1–10.
 9. Lu D, Danilowicz C, Tashjian TF, Prévost C, Godoy VG, Prentiss M. 2019. Slow extension of the invading DNA strand in a D-loop formed by RecA-mediated homologous recombination may enhance recognition of DNA homology. *Journal of Biological Chemistry* 294:8606–8616.
 10. Zhao Q, Feng Y, Zong Z. 2022. Conjugation of a Hybrid Plasmid Encoding Hypervirulence and Carbapenem Resistance in *Klebsiella pneumoniae* of Sequence Type 592. *Frontiers in Microbiology* 13:1–9.
 11. Craig NL. 1997. TARGET SITE SELECTION IN TRANSPOSITION. *Annual Review of Biochemistry* 66:437–474.
 12. Peters JE, Craig NL. 2001. Tn7: smarter than we thought. *Nature Reviews Molecular Cell Biology* 2:806–814.
 13. Roberts AP, Mullany P. 2009. A modular master on the move: the Tn916 family of mobile genetic elements. *Trends in Microbiology* 17:251–258.
 14. Hubbard ATM, Mason J, Roberts P, Parry CM, Corless C, van Aartsen J, Howard A, Bulgasim I, Fraser AJ, Adams ER, Roberts AP, Edwards T. 2020. Piperacillin/tazobactam resistance in a clinical isolate of *Escherichia coli* due to IS26-mediated amplification of *bla*TEM-1B. *Nature communications* 11:4915.
 15. Zhang P, Hu H, Shi Q, Sun L, Wu X, Hua X, McNally A, Jiang Y, Yu Y, Du X. 2023. The Effect of β -Lactam Antibiotics on the Evolution of Ceftazidime/Avibactam and Cefiderocol Resistance in KPC-Producing *Klebsiella pneumoniae*. *Antimicrobial Agents and*

Chemotherapy 67:2–10.

16. Schuster CF, Weber RE, Weig M, Werner G, Pfeifer Y. 2022. Ultra-deep long-read sequencing detects IS-mediated gene duplications as a potential trigger to generate arrays of resistance genes and a mechanism to induce novel gene variants such as bla CTX-M-243. *Journal of Antimicrobial Chemotherapy* 77:381–390.

At last, we appreciate for reviewers' warm work earnestly, and hope that the correction will meet with approval. Once again, thank you very much for your comments and suggestions.

Supplementary material for review_1

The results of quality assessment of the Illumina assembled data and average nucleotide identity analysis

Isolates ID	contigs	Largest contig	Total length	GC (%)	N50	ANI
kp1	123	333170	5781809	56.98	136470	98.8253
kp2	106	349308	5714432	57.07	151772	98.8769
kp3	117	355121	5801610	57.02	151772	98.8475
kp4	106	335012	5714782	57.07	176095	98.9228
kp5	115	355121	5805039	57.01	151632	98.8518
kp7	140	335012	5734875	57.04	176097	98.8812
kp8	109	334848	5717068	57.07	192089	98.8927
kp9	120	333171	5784066	56.99	136877	98.8662
kp10	108	335012	5713139	57.07	181893	98.9225
kp11	106	334706	5715015	57.07	181638	98.8912
kp12	98	355127	5641415	57.09	176091	98.8267
kp14	108	334808	5712132	57.07	151775	98.9243
kp15	96	365004	5642318	57.09	182680	98.8638
kp16	115	355121	5835045	56.9	151772	98.9002
kp17	105	334706	5715569	57.07	156677	98.9009
kp18	105	457044	5713771	57.07	176099	98.9238
kp19	100	355127	5706755	57.05	156679	98.9065
kp20	105	335012	5713268	57.07	181893	98.9099
kp21	85	1159560	5517940	57.21	185308	99.0097
kp22	101	365157	5707059	57.05	181639	98.8804
kp23	109	335012	5714767	57.07	170236	98.9077
kp24	106	335012	5715033	57.07	176095	98.8954
kp26	105	270340	5714812	57.07	176095	98.9057
kp28	108	335012	5713775	57.07	170236	98.9237
kp29	107	334950	5719866	57.07	181678	98.9273
kp30	103	355025	5642514	57.08	151819	98.879
kp31	96	354925	5643480	57.08	182680	98.8914
kp32	99	365004	5642897	57.08	176091	98.8511
kp33	103	335012	5714861	57.07	181893	98.8985
kp34	106	335012	5714468	57.07	181893	98.9204
kp35	103	355127	5700062	57.05	156679	98.8473
kp37	102	355127	5697952	57.05	170236	98.8474
kp38	106	335012	5714022	57.07	181892	98.903
kp39	126	332788	5784190	56.98	108676	98.8517
kp40	107	335012	5714147	57.07	156677	98.9231
kp41	107	335012	5713871	57.08	176095	98.9109
kp42	106	335012	5714683	57.07	176095	98.9181

kp46	112	335012	5712902	57.07	171908	98.9153
kp51	117	354815	5835149	56.9	151772	98.8603
kp53	103	355025	5706344	57.05	171742	98.8808
kp54	100	354923	5707046	57.05	176093	98.9169
kp55	109	334910	5712551	57.07	171886	98.9118
kp56	96	355127	5643798	57.09	192241	98.8615
kp59	109	335012	5714447	57.07	176095	98.8866
kp63	104	355127	5641495	57.09	181893	98.8844
kp66	122	340451	5782301	57.02	159822	98.8644
kp69	103	335012	5682249	57.07	182111	98.9342
kp70	99	355127	5640891	57.09	171886	98.8796
kp71	109	462647	5761473	56.95	182681	98.8647
kp73	111	335012	5713717	57.07	171061	98.863
kp80	113	335052	5718740	57.07	176135	98.8527
kp82	100	355127	5643619	57.09	170426	98.8603
kp84	99	365049	5708150	57.05	156678	98.9011
kp86	119	355121	5806131	57.01	159825	98.8459
kp87	116	365202	5814144	57.01	192281	98.823
kp88	107	352638	5741669	57.06	176137	98.8762
kp89	103	365196	5650466	57.08	183048	98.8602
kp91	109	335052	5724574	57.08	192237	98.9178
kp93	105	335052	5725962	57.06	176135	98.888
kp94	109	349850	5720340	57.07	176135	98.8956
kp97	105	355167	5704617	57.05	156718	98.8717
kp100	111	465870	5957816	57.05	182719	98.858
kp103	101	338457	5717051	57.07	196453	98.9306
kp106	101	365153	5649929	57.08	192281	98.8582
kp111	103	365196	5649647	57.08	192281	98.8558
kp116	105	355167	5649096	57.08	182719	98.8228
kp117	136	332679	5751645	57.04	189496	98.8737
kp118	121	335012	5754161	57.03	156677	98.9021
kp121	104	355167	5649701	57.08	192281	98.8667
kp122	123	365202	5815648	57	150817	98.8618
kp124	120	355121	5789270	57.03	159824	98.842
kp125	105	335052	5718693	57.07	192281	98.8966
kp126	111	334910	5713733	57.07	170236	98.9239
kp130	109	424040	5577542	57.19	203775	98.9156
kp131	105	355161	5749969	57.05	156719	98.8818
kp138	122	335436	5742961	57.05	176135	98.9161
kp139	108	335052	5717966	57.07	192281	98.917
kp142	87	392068	5598366	57.14	196434	98.9568

kp143	102	335052	5717729	57.07	176135	98.9306
kp144	76	1156152	5535531	57.19	247526	98.976
kp146	104	512055	5655336	57	152208	98.8765
kp148	105	424162	5577136	57.19	203899	98.8632
kp149	86	392072	5495865	57.37	157760	99.0055
kp151	101	365198	5712552	57.05	183048	98.8482
kp152	101	365155	5713291	57.05	156719	98.8506
kp153	105	357150	5706159	57.04	156719	98.8664
kp154	105	335052	5720310	57.07	192237	98.8994
kp156	114	355161	5810378	57	183146	98.8441
kp158	65	1159600	5478513	57.27	247566	99.0629
kp159	115	355379	5796326	57.01	151812	98.8601
kp160	116	365197	5842975	56.9	151812	98.8716
kp161	146	365197	5988861	56.58	128748	98.8978
kp162	120	335052	5727702	57.07	162980	98.9353
kp164	111	335052	5697470	57.09	176135	98.8926
kp165	108	457084	5696471	57.09	176135	98.9316
kp166	133	333226	5946012	56.63	129437	98.9073
kp167	107	335009	5696701	57.1	164736	98.9154
kp168	110	335052	5696662	57.09	176135	98.904
kp169	133	353908	5800451	57	176131	98.8582
kp171	111	335433	5697016	57.09	156717	98.8867
kp173	108	335436	5703191	57.09	176135	98.9042
kp174	117	365197	5752464	57.08	160184	98.8607
kp175	120	335052	5727224	57.07	162980	98.9162
kp176	117	365197	5752634	57.08	160184	98.8877
kp177	137	353908	5740405	57.04	183048	98.881
kp178	114	474692	5752837	57.08	160184	98.8709
kp179	134	246004	5707958	57.08	105200	98.9344
kp180	106	451546	5737674	57.06	146891	98.9056
kp181	122	457084	5741817	57.05	176135	98.9383
kp182	108	335436	5696743	57.09	176135	98.8906
kp183	131	365152	5740442	57.08	183048	98.9257
kp184	111	335052	5696674	57.09	176135	98.887
kp185	117	355161	5751572	57.08	151812	98.8489
KP186	144	332932	5987803	56.58	129477	98.9021
kp187	119	365197	5752326	57.08	141154	98.8567
kp188	140	430086	5987729	56.58	176019	98.8912
kp189	115	340460	5727302	57.05	183048	98.9072
kp190	118	365154	5752811	57.08	160184	98.8521
kp192	121	365154	5752692	57.08	143755	98.9032

kp193	110	334971	5696546	57.09	176023	98.8845
kp194	134	365197	5969035	56.59	136023	98.9072
kp195	111	335009	5690943	57.1	176095	98.9335
kp196	140	365197	5954377	56.63	136023	98.914
kp197	141	332901	5955556	56.63	129477	98.9092
kp198	108	335052	5696497	57.09	176135	98.871
kp199	112	335052	5697423	57.09	164776	98.9368
kp200	109	335052	5696352	57.09	176023	98.8782
kp201	109	335052	5696983	57.09	176135	98.8968
kp202	112	335436	5698224	57.09	164773	98.9116
kp203	144	332901	5955310	56.63	129865	98.8975
kp204	142	365153	5987712	56.58	127920	98.8924
kp207	134	429396	5942023	56.64	136023	98.9098
kp208	120	457084	5742045	57.05	192237	98.9079
kp209	127	365196	5777668	57	138850	98.8624
kp211	101	451546	5733843	57.07	146894	98.9036
kp212	104	335052	5696898	57.09	176023	98.8942
kp213	103	457084	5693911	57.1	196453	98.9292
kp214	105	338335	5693337	57.1	196453	98.9125
kp215	128	328679	5798392	56.97	102429	98.8292
kp222	105	338417	5686697	57.1	176095	98.9089
kp226	141	365153	5954451	56.63	136023	98.8791
kp227	110	335052	5702744	57.09	164776	98.9163
kp228	108	365047	5700443	57.03	171908	98.8501
kp231	109	335052	5695903	57.1	164820	98.9093
kp232	119	335052	5733938	57.05	176135	98.8948
kp236	96	512056	5655070	57	196449	98.9084
kp237	126	335012	5727546	57.07	151772	98.8769
kp239	124	355121	5809677	57	143076	98.8778
kp241	114	335012	5701051	57.09	176095	98.9073
kp242	126	396081	5827648	57.04	151811	98.9068
kp245	139	365196	5801510	57	183048	98.9097
kp246	108	338456	5717943	57.08	176135	98.8857
kp248	106	365153	5649585	57.08	182719	98.8566
kp250	56	1159600	5370352	57.37	270420	99.1241
kp252	111	335012	5714027	57.07	181893	98.8947
kp253	118	335052	5705603	57.09	176135	98.9316
kp255	110	338417	5688103	57.1	151819	98.8832
kp256	124	396496	5825041	57.04	192238	98.8576
kp258	108	355127	5641596	57.09	151775	98.8591
kp259	100	512016	5651183	57.01	152353	98.8965

kp262	102	512056	5654962	57	192212	98.9235
kp263	102	512016	5646261	57.01	171886	98.8703
kp264	114	335012	5714580	57.09	156678	98.9099
kp266	104	338457	5674513	57.1	164772	98.8985
kp267	106	451546	5654146	57.11	146891	98.8981
kp268	103	355127	5707263	57.03	182680	98.839
kp270	62	665997	5342081	57.4	270340	99.1353
kp273	123	333226	5819009	57.04	151771	98.9127
kp276	121	355121	5805264	57.01	170236	98.8543
kp278	126	469972	5822052	57.04	151771	98.9379
kp283	124	332783	5791906	56.99	141464	98.8556
kp284	120	333171	5795152	57.06	147569	98.9151
kp286	123	355059	5797753	57.02	183146	98.8712
kp287	120	353868	5762806	57	152437	98.8596
kp288	124	396215	5828188	57.03	151811	98.872
kp289	127	396214	5828005	57.03	151811	98.894
kp290	120	396358	5800642	57.07	190778	98.9216
KP293	107	353664	5765875	56.99	181893	98.8695
KP294	108	353766	5765159	56.99	182679	98.9101
KP295	119	474652	5806801	57.01	151819	98.8631
KP298	111	365003	5765285	56.99	176093	98.9115
KP299	107	353725	5770109	56.99	151812	98.8705
KP300	133	365201	5866813	56.86	141504	98.8463
KP302	110	365152	5772605	56.99	182719	98.859
KP303	123	474652	5811702	57	159825	98.8749
KP305	94	512016	5647499	57.01	153184	98.906
KP312	101	510792	5646808	57.01	181893	98.8891
KP313	82	392032	5495123	57.36	160275	99.0166
KP316	95	371934	5628505	57.02	192197	98.8896
KP318	110	332885	5795018	56.97	143746	98.8811
KP319	132	353664	5809554	56.98	181892	98.8518
KP320	112	353664	5763661	57	181893	98.8733
KP325	127	365207	5812501	57.01	159863	98.839
KP335	112	353868	5761949	57	181893	98.8703
KP337	111	365155	5762413	57	181893	98.8906
KP344	122	355161	5803173	57.01	182817	98.8679
KP345	118	364862	5828745	56.98	181893	98.8802
KP347	123	332776	5794227	56.98	141464	98.8875
KP348	120	364915	5793219	56.98	141464	98.8464
KP350	122	365011	5849082	56.95	192281	98.8792
KP351	122	355121	5838766	56.97	181893	98.8984

KP353	57	907799	5366228	57.37	247526	99.1122
KP354	111	353806	5772897	56.99	182719	98.8603
KP355	137	353908	5823588	56.97	182719	98.8638
KP357	108	364865	5762898	57	181893	98.8808
KP364	135	365196	5819129	56.98	183048	98.8868
KP366	114	365156	5763925	57	181893	98.8955
KP368	114	365196	5770237	57	176131	98.8195
KP372	61	1076973	5366313	57.37	247526	99.0858
kp373	122	289905	5820009	56.99	119536	98.8766
KP374	106	353868	5763679	57	176093	98.8641
KP376	117	355121	5789901	57.03	170426	98.8528
KP377	108	353868	5763050	57	182680	98.8752
kp378	117	365542	5807833	57.02	152437	98.8504
KP379	110	364865	5760958	57	176093	98.8643
KP380	109	365003	5765096	57	182681	98.8374
KP385	106	353868	5764952	56.99	181893	98.9311
KP393	82	568057	5566823	57.12	203859	99.0414
KP394	120	365054	5796479	56.98	141464	98.8801
KP397	115	288792	5831142	56.98	124687	98.8347
KP411	120	364919	5804398	57.01	163708	98.8563
KP413	81	392032	5389712	57.35	176093	98.9735
KP421	82	392032	5492533	57.36	176093	99.0252
KP424	116	355121	5798977	57.01	151772	98.8665
KP426	109	289766	5794100	57	116841	98.8728
KP428	116	364920	5801920	57.02	140217	98.8935
KP430	77	392067	5477864	57.34	203775	98.9836
KP431	79	391931	5500261	57.35	196434	98.9864
KP434	115	365202	5808133	57.01	159866	98.8465
KP436	116	365158	5811093	57.01	192281	98.8485
KP438	121	469286	5811392	57	159826	98.8647
KP439	114	482255	5833791	57	140931	98.898
KP442	106	289795	5793988	57	124651	98.9085
KP443	107	289796	5802045	57	116797	98.9214
KP445	109	355121	5804910	57.02	181893	98.8575
KP446	111	289796	5794787	57	116841	98.8928
KP447	107	289658	5789278	57.01	116797	98.9121

Supplementary material for review_2

Antibiotic (β -lactam or β -lactam/lactamase inhibitor combinations) usage of patients colonized by KPC-2-CRKP during the screening period.

	piperacillin/tazobactam	cefoperazone/sulbactam	imipenem	meropenem	biapenem	others
P1	N	N	0.5g q6h 19days	N	N	N
P2	4.5g q8h 19days	N	N	N	N	N
P3	N	N	0.5g q8h 2days	N	N	N
P4	N	N	0.5g q8h 3days	N	N	ceftazidime/avibactam 2.5g q8h 1day
P5	N	2g q12h 22days	N	N	N	N
P6	N	N	N	0.5g q8h 4days	N	cefmetazole 2g q12h 2days
P7	4.5g q8h 4days	N	N	N	N	N
P8	4.5g q8h 1day	2g q8h 25days	1g q12h 1days	1g q8h 24days	N	ceftriaxone 2g qd 1day
P9	N	N	1g q12h 43days	1g q8h 3days	N	N
P10	4.5g q8h 3months	N	N	N	N	N
P11	N	2g q8h 3days	0.5g q12h 35days	N	0.3g q6h 14days	N
P12	4.5g q8h 14days; 2.25g q8h 11days	2g q8h 16days	N	1g q8h 4days	N	N
P13	2.25g q8h 3days	N	N	N	N	N
P14	2.25g q8h 26days	N	N	N	N	cefuroxime 1.5g qd 1day
P15	N	N	1g q12h 71days	N	N	N
P16	4.5g q8h 22days	N	N	N	0.3g q12h 3days	N
P17	4.5g q8h 1day	2g q8h 82days	N	N	N	N
P18	4.5g q8h 1day	N	0.5g q12h 19days	N	N	N
P19	4.5g q8h 32days	2g q8h 17days	N	N	N	cefmetazole 2g q12h 1day
P20	4.5g q8h 2days	2g q8h 5days	N	N	N	N

P21	4.5g q8h 9days	N	N	1g q8h 3days	N	N
P22	N	2g q8h 1day	N	0.5g q8h 30days	N	N
P23	4.5g q8h 5days	2g q12h 19days	0.5g q8h 3days	N	N	cefmetazole 2g q12h 1day; ceftazidime/avibactam 2.5g q8h 18days
P24	N	N	N	N	N	N
P25	4.5g q8h 12days	2g q8h 13days	N	N	N	N
P26	4.5g q8h 6days	N	N	N	N	N
P27	N	2g q8h 19days	0.5g q6h 16days	1g q8h 4days	N	ceftazidime/avibactam 2.5g q8h 46days
P28	4.5g q8h 11days	N	N	N	N	N
P29	N	2g q8h 3days	N	N	N	N
P30	N	N	0.5g q8h 3days; 0.25g q8h 2days	N	N	N
P31	N	2g q12h 27days	N	N	N	N
P32	4.5g q8h 29days	N	N	N	N	N
P33	4.5g q8h 1day	N	N	1g q8h 5days	N	N
P34	N	N	0.5g q12h 2days; 0.25g q12h 5days	N	N	N
P35	N	2g q6h 2days	N	N	N	N
P36	N	N	0.5g q6h 11days; 0.5g q8h 10days	0.5g q8h 2days; 1g q8h 1day	N	N
P37	4.5g q8h 37days	N	0.5g q8h 4days	N	N	cefmetazole 2g q12h 1day
P38	4.5g q8h 1day	N	0.5g q6h 6days	N	N	N

P39 4.5g q8h 3days

2g q12h 5days

N

N

N

N

N: not used

Re: mSystems00924-23R1 (Genome sequencing unveils bla_{KPC-2}-harboring plasmids as drivers of enhanced resistance and virulence in nosocomial *Klebsiella pneumoniae*)

Dear Prof. Yunsong Yu:

Your manuscript has been accepted, and I am forwarding it to the ASM production staff for publication. Your paper will first be checked to make sure all elements meet the technical requirements. ASM staff will contact you if anything needs to be revised before copyediting and production can begin. Otherwise, you will be notified when your proofs are ready to be viewed.

Featured Image Submissions: If you would like to submit a potential Featured Image, please email a file and a short legend to mSystems@asmusa.org. Please note that we can only consider images that (i) the authors created or own and (ii) have not been previously published. By submitting, you agree that the image can be used under the same terms as the published article. File requirements: square dimensions (4" x 4"), 300 dpi resolution, RGB colorspace, TIF file format.

Sincerely,
Sima Tokajian
Editor
mSystems

Reviewer #1 (Comments for the Author):

The authors have addressed my comments. Congratulations on all their hard work.

Reviewer #2 (Comments for the Author):

Authors provided adequate responses to my concerns.

Reviewer #3 (Comments for the Author):

None